# GmNet: Revisiting Gating Mechanisms From A Frequency View

**Yifan Wang**[1]   **Xu Ma**[1]   **Yitian Zhang**[1]   **Yizhou Wang**[1]   **Zhongruo Wang**[2]   **Sung-Cheol Kim**[3]
**Vahid Mirjalili**[3]   **Vidya Renganathan**[3]   **Yun Fu**[1]
[1] Northeastern University     [2] UC Davis     [3] FM-Global
{wang.yifan25, ma.xu1, zhang.yitian}@northeastern.edu, yunfu@ece.neu.edu

## Abstract

Lightweight neural networks, essential for on-device applications, often suffer from a low-frequency bias due to their constrained capacity and depth. This limits their ability to capture the fine-grained, high-frequency details (e.g., textures, edges) that are crucial for complex computer vision tasks. To address this fundamental limitation, we perform the first systematic analysis of gating mechanisms from a frequency perspective. Inspired by the convolution theorem, we show how the interplay between element-wise multiplication and non-linear activation functions within Gated Linear Units (GLUs) provides a powerful mechanism to selectively amplify high-frequency signals, thereby enriching the model's feature representations. Based on these findings, we introduce the Gating Mechanism Network (GmNet), a simple yet highly effective architecture that incorporates our frequency-aware gating principles into a standard lightweight backbone. The efficacy of our approach is remarkable: without relying on complex training strategies or architectural search, GmNet achieves a new state-of-the-art for efficient models. Our code can be found in https://github.com/YFWang1999/GmNet

## 1 Introduction

Designing neural networks that are both highly accurate and computationally efficient is a central challenge in modern vision task. Lightweight models are essential for on-device applications, but their reduced capacity often limits their ability to capture the fine-grained details necessary for complex recognition tasks. A growing body of research suggests this limitation stems from a spectral bias, where standard neural network architectures preferentially learn simple, low-frequency global patterns while struggling to capture high-frequency information corresponding to textures and edges Rahaman et al. (2019); Tancik et al. (2020). This fundamental performance gap motivates the exploration of architectural innovations that can improve a model's representational power without sacrificing efficiency. This bias is particularly pronounced in efficient models whose limited capacity hinders their ability to learn complex, high-frequency information. This limitation motivates our analysis of Gated Linear Units (GLUs)—a computationally inexpensive mechanism already proven effective in various high-performance models De et al. (2024); Liu et al. (2021); Gu & Dao (2023). While their success is often attributed to adaptive information control, their impact on a network's spectral properties remains largely unexplored. We hypothesize that the element-wise multiplication at the core of GLUs, which corresponds to convolution in the frequency domain, provides a direct mechanism to modulate this spectral bias and enrich a model's high-frequency learning.

To build intuition, we present an example that visually illustrates how GLUs alter a network's response to different frequency components of an image as shown in Fig. 1. We take a standard convolution-based lightweight building block (the top one) and create a variant by incorporating our proposed gating unit ( the bottom one). We first provide an input image decomposed into different frequency components from low to high. The visualizations show that the baseline model primarily performs accurate on the low-frequency information, struggling capturing crucial textural details which leads to an incorrect classification on the raw image. In sharp contrast, the model with GLU demonstrates a more balanced spectral response, effectively learning from both low and high-frequency components to form a richer representation. This simple experiment provides initial

illustration that gating mechanisms can directly counteract the low-frequency bias in many efficient architectures.

The mechanism enabling this enhanced spectral response is rooted in the convolution theorem: element-wise multiplication in the spatial domain is equivalent to convolution in the frequency domain. This operation allows the network to create complex interactions between different frequency bands, enriching the feature hierarchy. However, as prior work has noted Wang et al. (2020); Yin et al. (2019), naively amplifying high frequencies can make a model overly sensitive to noise. The key, therefore, is selective modulation. We contend that Gated Linear Units, by pairing the multiplication with a data-dependent gate and a non-linear activation function, provide exactly this control. They allow the model to learn when to integrate high-frequency details and how much to trust them, effectively amplifying useful signals while remaining robust to high-frequency noise.

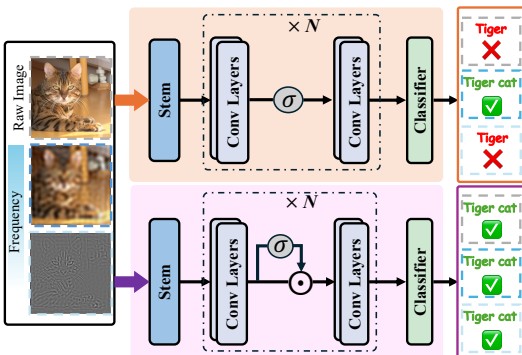

To put these principles into practice, we introduce the Gating Mechanism Network (GmNet), a lightweight architecture designed to leverage the spectral advantages of gating. By effectively capturing information across the full frequency spectrum, GmNet demonstrates that a structurally-motivated design can lead to substantial practical gains. Compared to the existing methods Ma et al. (2024a;b) which also involve gating designs, our design leverages a self-reinforcing gating mechanism in which the modulation and gating signals are derived from a shared representation. This alignment ensures that salient variations, particularly those associated with high-frequency components, are consistently emphasized rather than suppressed. In contrast, methods based on independent projections often act as generic filters, leading to

Figure 1: An illustration of how GLUs affect neural networks in classifying different frequency parts of an image. $\sigma$ means activation function. Starting with a raw image of a 'Tiger cat', we break it down into different frequency bands. The lowest frequency shows a recognizable outline, the higher frequency retains the general shape of the cat, but the highest frequency is almost unrecognizable. Predictions of different components are given in the left of different models. This example demonstrates two points: 1. Although low-frequency decomposed images closely resemble the originals, accurate recognition of it does not guarantee accurate recognition of the original images, and 2. GLUs improve the NNs' ability to learn higher frequency components effectively.

weaker sensitivity to subtle variations that are critical for classification. Consequently, our approach is inherently more effective in preserving and enhancing high-frequency information. The results are compelling: without relying on advanced training techniques, our GmNet-S3 model achieves 81.3% top-1 accuracy on ImageNet-1K. This surpasses EfficientFormer-L1 by a significant 4.0% margin while simultaneously being 4x faster on an A100 GPU, showcasing a new state-of-the-art in efficient network design.

We summarize the key contributions of this work as follows: (1) We provide the first systematic analysis of Gated Linear Units (GLUs) from a frequency perspective, establishing a clear link between their core operations and their ability to modulate a network's spectral response. (2) We demonstrate that this spectral modulation can directly counteract the inherent low-frequency bias in many lightweight architectures, enabling them to learn more balanced and detailed feature representations from both low and high frequencies. (3) Based on these insights, we introduce the Gating Mechanism Network (GmNet), a simple yet powerful lightweight architecture that achieves a new state-of-the-art in performance and efficiency, validating the practical benefits of our frequency-based design principles.

## 2 RELATED WORK

**Gated Linear Units.** The Gated Linear Unit (GLU) Dauphin et al. (2017), and its modern variants like SwiGLU Shazeer (2020), have become integral components in state-of-the-art deep learning

models. Originally developed for sequence processing, their ability to selectively control information flow with minimal computational overhead has led to widespread adoption. In Natural Language Processing, they are central to powerful Transformers such as Llama3 Dubey et al. (2024) and state-space models like Mamba Gu & Dao (2023), where they are lauded for improving training dynamics. This empirical success has spurred their integration into computer vision architectures; models like gMLP Liu et al. (2021) have shown that replacing self-attention with simple gating-MLP blocks can yield competitive performance. However, the prevailing understanding of these mechanisms remains largely functional—they are viewed as adaptive 'information gates.' A critical gap exists in the analysis of their impact on a network's fundamental learning properties. Specifically, no prior work has systematically analyzed GLUs from a frequency perspective or connected their operational mechanism to the well-documented problem of low-frequency bias in vision models.

**Frequency Learning.** Analyzing neural networks from a frequency perspective has revealed a fundamental learning dynamic known as spectral bias: networks of various types consistently learn simple, low-frequency patterns much faster than complex, high-frequency details Rahaman et al. (2019); Yin et al. (2019); Tancik et al. (2020). While initially explored in regression tasks, this bias presents a significant bottleneck for image classification, particularly in lightweight models. Due to their constrained capacity, these models struggle to capture the high-frequency information corresponding to textures and edges, limiting their overall performance. Furthermore, the use of high-frequency components involves a delicate trade-off; while they are critical for accuracy, they can also make models more susceptible to high-frequency noise, impacting robustness Wang et al. (2020). Crucially, while prior work has adeptly characterized these phenomena, it has largely focused on analysis and diagnosis. A clear gap remains in proposing and studying specific, efficient architectural mechanisms that can actively manage this accuracy-robustness trade-off and explicitly counteract spectral bias within a model's design.

**Lightweight Networks.** The design of lightweight networks has predominantly followed two streams: pure convolution-based architectures like MobileOne Vasu et al. (2023b) and RepVit Wang et al. (2024), and hybrid approaches incorporating self-attention, such as EfficientFormerV2 Li et al. (2022). While these lines of work have successfully pushed the frontiers of computational efficiency, they are built upon operations that are now understood to have a strong intrinsic low-frequency bias Tang et al. (2022); Bai et al. (2022). This foundational bias is often exacerbated in the lightweight regime; the aggressive optimization for fewer parameters and lower FLOPs further restricts a model's capacity to learn essential high-frequency information. Consequently, the current paradigm for efficient network design contains a significant blind spot: it has optimized for computational metrics while largely overlooking the spectral fidelity of the learned representations. This leaves a clear opening for new design principles that explicitly aim to correct this low-frequency bias from the ground up.

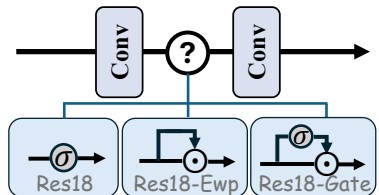

Figure 2: Block design of different variants of ResNet18 where $\odot$ represents the element-wise product and $\sigma$ means the activation function.

## 3 REVISITING GATING MECHANISMS FROM A FREQUENCY VIEW

We begin by defining the components associated with different frequency bands and outlining the details of our experimental setup. With decomposing the raw data $\mathbf{z}$ into the high-frequency part $\mathbf{z}_h$ and the low-frequency part $\mathbf{z}_l$ where $\mathbf{z} = \mathbf{z}_h + \mathbf{z}_l$. We partition the spectrum into two complementary components based on a predefined cutoff radius $r$. Specifically, the frequency representation $z$ is decomposed into a low-frequency part $z_l$ and a high-frequency part $z_h$, satisfying

$$z = z_l + z_h, \quad (z_l, z_h) = \Theta(z; r), \tag{1}$$

where $\Theta(\cdot; r)$ represents a radial masking operator that separates spectral coefficients according to their distance from the origin in the frequency plane. In this formulation, coefficients located within radius $r$ are assigned to $z_l$, while those outside the radius are assigned to $z_h$. This decomposition enables us to analyze model behaviors under controlled spectral partitions.

We select three vision backbones including ResNet-18 He et al. (2016), MobileNetv2 Sandler et al. (2018) and EfficientFormer-v2 Li et al. (2023) as representations to demonstrate the drawbacks of

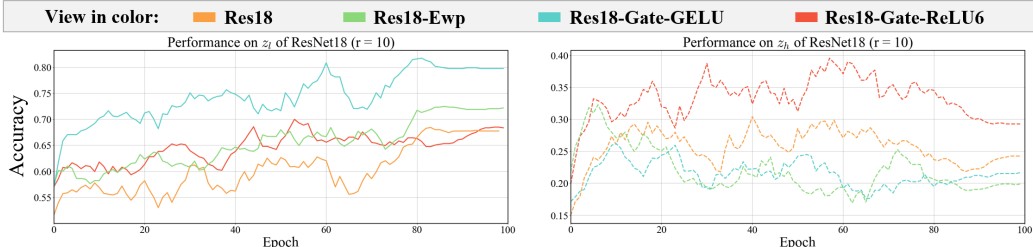

Figure 3: Comparison among Res18, Res18-Ewp, Res18-Gate-ReLU6 and Res18-Gate-GELU. The $r$ represents the threshold of determining the boundary between low-frequency and high-frequency. We plot the learning curves of Resnet18 and its variants for 100 epochs, together plotted with the accuracy of different frequency components $z_i$. We set $r$ to 10. All curves of $z$ are from the test set. The legends can be found in the top of the figure. We also provide more results with different $r$ and different settings in the appendix.

the CNN networks and transformer-based architectures on capturing the high frequency information and how the gating mechanism improve the capability of learning high-frequency components. Modifications to the network blocks of ResNet-18 are depicted in Fig. 2. We evaluate the classification performance on different frequency components of the input images at each training epoch. Changes in accuracy over time provide insights into the learning dynamics within the frequency domain Wang et al. (2020). To avoid the occasionality, we calculated the average over three training runs.

## 3.1 EFFECT OF ELEMENT-WISE PRODUCT

Inspired by the *convolution theorem*, we first give an insight into why element-wise product can encourage NNs to learn on various frequency components from a frequency view. The convolution theorem states that for two functions $u(x)$ and $v(x)$ with Fourier transforms $U$ and $V$,

$$(u \cdot v)(x) = \mathcal{F}^{-1}(U * V), \tag{2}$$

where $\cdot$ and $*$ denote element-wise multiplication and convolution respectively, and $\mathcal{F}$ is the Fourier transform operator defined as $\mathcal{F}[f(t)] = F(\omega) = \int_{-\infty}^{+\infty} f(t)e^{-j\omega t}\,dt$. This indicates that element-wise multiplication in the spatial domain corresponds to convolution in the frequency domain.

To see its implication more clearly, consider the simplest situation: the self-convolution of a function. If the support set of $\mathcal{F}(\omega)$ is $[-\Omega, \Omega]$, then the support set of $\mathcal{F} * \mathcal{F}(\omega)$ will expand to $[-2\Omega, 2\Omega]$. In other words, self-convolution broadens the frequency spectrum. With this enriched frequency content, neural networks have more opportunities to capture and learn from both high-frequency and low-frequency components.

## 3.2 HOW ACTIVATION FUNCTION WORKS?

We begin by analyzing how an activation function's smoothness influences the frequency characteristics of the features it produces. There is a well-established principle in Fourier analysis that connects a function's smoothness to the decay rate of its Fourier transform's magnitude. For a function $f(t)$ that is sufficiently smooth (i.e., its $n$-th derivative $f^{(n)}(t)$ exists and is continuous), the magnitude of its Fourier transform, $|F(\omega)|$, is bounded and decays at a rate proportional to $1/|\omega|^n$ for large $\omega$. This is a direct consequence of the differentiation property of the Fourier transform:

$$\mathcal{F}[f^{(n)}(t)] = (j\omega)^n F(\omega), \tag{3}$$

This property implies that the smoother a function is (i.e., the more continuous derivatives it has), the more rapidly its high-frequency components decay.

Conversely, functions with discontinuities or sharp "corners" where derivatives are undefined (such as the kink in ReLU-like activations) are known to possess significant high-frequency energy. These

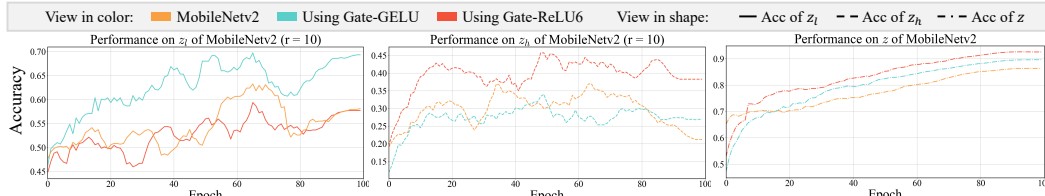

Figure 4: Comparison among different variants of MobileNetv2. Different architectures respond differently to specific frequency component. To ensure an informative comparison, we select representative frequency thresholds tailored to each model where we set $r$ to 10. Additional results under other threshold configurations and other based models are included in the supplementary material.

sharp features require a broad spectrum of high-frequency sinusoids to be accurately represented. This leads to a Fourier transform that decays much more slowly. For example, a function with a simple discontinuity will have a spectrum that decays only at a rate of $1/|\omega|$. Therefore, we hypothesize that non-smooth activation functions will encourage the network to retain and utilize more high-frequency information compared to their smooth counterparts like GELU and Swish, which is infinitely differentiable.

To validate this hypothesis, we conduct an experiment to compare the frequency learning of a representative smooth activation (GELU) against a non-smooth one (ReLU6) within a ResNet18 architecture. As shown in Fig. 3, the model using the non-smooth ReLU6 activation consistently outperforms the GELU variant in learning from high-frequency components across different thresholds. This result supports our hypothesis, illustrating a clear practical difference between these two activation types. The superior performance of ReLU6 on high-frequency data suggests that the slow spectral decay associated with non-smooth functions can be beneficial for tasks requiring fine-grained detail. Conversely, the GELU variant shows a stronger relative performance on low-frequency components, indicating its suitability for capturing broad, structural patterns. While a more exhaustive study is needed, this experiment provides clear evidence for the link between activation smoothness and a model's spectral learning preferences.

## 4 GATING MECHANISM NETWORK (GMNET)

### 4.1 RETHINKING CURRENT LIGHTWEIGHT MODEL ARCHITECTURES FROM A FREQUENCY PERSPECTIVE

Before introducing our proposed network, we first investigate the importance of capturing high-frequency information in efficient architectures by modifying existing efficient models to incorporate Gated Linear Units (GLUs). Specifically, we select one representative architecture: the pure CNN-based MobileNetV2 Sandler et al. (2018). We replace the activation functions in their MLP blocks with a simple GLU. Detailed architectural modifications are provided in the appendix.

As shown in Fig. 4, we present the testing accuracy curves under the frequency threshold $r = 10$. Our results demonstrate that integrating GLUs improves classification accuracy on high-frequency components. Notably, this improvement in high-frequency classification also correlates with a gain in overall performance. Furthermore, we observe that using GELU as the activation function within the GLUs enhances performance on low-frequency components, though it has a relatively minor effect on overall accuracy. These findings suggest that effectively modeling high-frequency information is more crucial for improving the performance of lightweight neu-

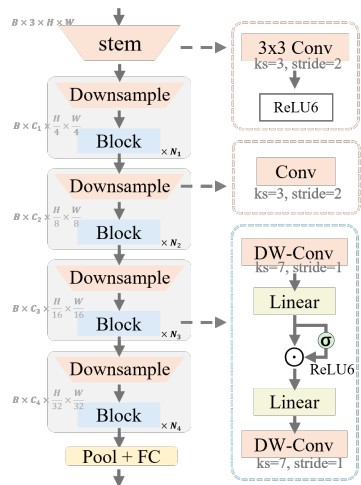

Figure 5: **GmNet architecture**. Gm-Net adopts a traditional hybrid architecture, utilizing convolutional layers to down-sample the resolution and double the number of channels at each stage.

ral networks. It underscores the critical role of frequency-aware design in the lightweight networks. Moreover, we also conduct similar experiments on the transformer-based model EfficientFormer-V2 Li et al. (2023) which can be found in the appendix.

## 4.2 ARCHITECTURE OF GMNET

To address the limitation of low-frequency bias for current lightweight network designs, our proposed method named as GmNet integrates a simple gated linear unit into the block as illustrated in Fig. 5. GmNet offers both conceptual and practical advantages on encouraging the model to learn from a broader range of frequency regions, especially the high-frequency domain.

GmNets employ an extremely streamlined model architecture, carefully designed to minimize both parameter count and computational speed, making them particularly suitable for deployment in resource-constrained environments. We incorporate two depth-wise convolution layers with kernel sizes of $7 \times 7$ at the beginning and end of the block respectively to facilitate the integration of low- and high-frequency information. At the core of the block, we have two $1 \times 1$ convolution layers and a simple gated linear unit. We use the ReLU6 as the activation function.

GmNet uses a simplified GLU structure for two reasons: (1) to keep the model as lightweight as possible, reducing computational load; and (2) ensuring that high-frequency signals can be better enhanced without adding any additional convolutional or fully connected layers within the GLU. Furthermore, our gate unit is more interpretable, aligning with our analysis of GLUs in the frequency domain. Experimental results and ablation studies consistently demonstrate the superiority of our model, validating its design in accordance with our GLU frequency domain studies. We also show that the simplest structure achieves the optimal trade-off between efficiency and effectiveness.

## 5 EXPERIMENTS

In this section, we provide extensive experiments to show the superiority of our model and ample ablation studies to demonstrate the effectiveness of components of our method.

### 5.1 RESULTS IN IMAGE CLASSIFICATION

**Experimental Setup.** We evaluate our models on the ImageNet-1K benchmark using an input resolution of $224 \times 224$ for both training and inference. To construct different GmNet variants, we vary the stage depth, embedding width, and expansion ratio while preserving the overall architectural design. Detailed configurations of each model scale are provided in the appendix. All networks are optimized from random initialization for 300 epochs using the AdamW optimizer with an initial learning rate of $3 \times 10^{-3}$. The global batch size is set to 2048. Additional training hyperparameters, including data augmentation and regularization settings, are summarized in the supplementary material. For latency evaluation, we export the trained PyTorch models to ONNX format and measure inference time on both an NVIDIA A100 GPU and a mobile device (iPhone 14). To ensure fair deployment comparison, the mobile benchmarks are conducted via CoreML conversion. Importantly, no re-parameterization, distillation, or architecture search techniques are applied; all reported results correspond to standard end-to-end training.

Table 1: Comparison of Efficient Models on ImageNet-1k. Latency is evaluated across various platforms.

| Model | Top-1 | Params | FLOPs | Latency (ms) | |
|---|---|---|---|---|---|
| | (%) | (M) | (G) | GPU | Mobile |
| FasterNet-T0 Chen et al. (2023) | 71.9 | 3.9 | 0.3 | 2.5 | 0.7 |
| MobileV2-1.0 Sandler et al. (2018) | 72.0 | 3.4 | 0.3 | 1.7 | 0.9 |
| ShuffleV2-1.5 Ma et al. (2018) | 72.6 | 3.5 | 0.3 | 2.2 | 1.3 |
| EfficientFormerV2-S0 Li et al. (2023) | 73.7 | 3.5 | 0.4 | 2.0 | 0.9 |
| MobileNetv4-Conv-S Qin et al. (2024) | 73.8 | 3.8 | 0.2 | 2.2 | 0.9 |
| StarNet-S2 Ma et al. (2024a) | 74.8 | 3.7 | 0.5 | 1.9 | 0.9 |
| LSNet-T Wang et al. (2025) | 74.9 | 11.4 | 0.3 | 2.9 | 1.8 |
| GmNet-S1 | **75.5** | 3.7 | 0.6 | **1.6** | 1.0 |
| EfficientMod-xxs Ma et al. (2024b) | 76.0 | 4.7 | 0.6 | 2.3 | 18.2 |
| Fasternet-T1 Chen et al. (2023) | 76.2 | 7.6 | 0.9 | 2.5 | 1.0 |
| EfficientFormer-L1 Li et al. (2022) | 77.2 | 12.3 | 1.3 | 12.1 | 1.4 |
| StarNet-S3 Ma et al. (2024a) | 77.3 | 5.8 | 0.7 | 2.3 | 1.1 |
| MobileOne-S2 Vasu et al. (2023b) | 77.4 | 7.8 | 1.3 | 1.9 | 1.0 |
| RepViT-M0.9 Wang et al. (2024) | 77.4 | 5.1 | 0.8 | 3.0 | 1.1 |
| EfficientFormerV2-S1 Li et al. (2023) | 77.9 | 4.5 | 0.7 | 3.4 | 1.1 |
| GmNet-S2 | **78.3** | 6.2 | 0.9 | **1.9** | 1.1 |
| EfficientMod-xs Ma et al. (2024b) | 78.3 | 6.6 | 0.8 | 2.9 | 22.7 |
| StarNet-S4 Ma et al. (2024a) | 78.4 | 7.5 | 1.1 | 3.3 | 1.1 |
| SwiftFormer-S Shaker et al. (2023) | 78.5 | 6.1 | 1.0 | 3.8 | 1.1 |
| RepViT-M1.0 Wang et al. (2024) | 78.6 | 6.8 | 1.2 | 3.6 | 1.1 |
| UniRepLKNet-F Ding et al. (2024) | 78.6 | 6.2 | 0.9 | 3.1 | 3.5 |
| GmNet-S3 | **79.3** | 7.8 | 1.2 | 2.1 | 1.3 |
| RepViT-M1.1 Wang et al. (2024) | 79.4 | 8.3 | 1.3 | 5.1 | 1.2 |
| MobileOne-S4 Vasu et al. (2023b) | 79.4 | 14.8 | 2.9 | 2.9 | 1.8 |
| FastViT-S12 Vasu et al. (2023a) | 79.8 | 8.8 | 1.8 | 5.3 | 1.6 |
| MobileNetv4-Conv-M Qin et al. (2024) | 79.9 | 9.2 | 1.0 | 9.2 | 1.4 |
| LSNet-B Wang et al. (2025) | 80.3 | 23.2 | 1.3 | 6.2 | 3.6 |
| EfficientFormerV2-S2 Li et al. (2023) | 80.4 | 12.7 | 1.3 | 5.4 | 1.6 |
| EfficientMod-s Ma et al. (2024b) | 81.0 | 12.9 | 1.4 | 4.5 | 35.3 |
| RepViT-M1.5 Wang et al. (2024) | 81.2 | 14.0 | 2.3 | 6.4 | 1.7 |
| LeViT-256 Graham et al. (2021) | 81.5 | 18.9 | 1.1 | 6.7 | 31.4 |
| GmNet-S4 | **81.5** | 17.0 | 2.7 | **2.9** | 1.9 |

**Compared with the state-of-the-art.** The experimental results are presented in Table 1. Without any strong training strategy, GmNet delivers impressive performance compared to many state-of-the-art lightweight models. With a comparable latency on GPU, GmNet-S1 outperforms MobileV2-1.0 by 3.5%. Notably, GmNet-S2 achieves 78.3% with only 1.9ms on the A100 which is a remarkable achievements for the models under 1G FLOPS. GmNet-S3 outperforms RepViT-M1.0 and StarNet-S4 by 1.9% and 0.9% in top-1 accuracy with 1.1 ms and 1.4 ms faster on the GPU latency, respectively. The improvements on the speed are over 30%. Additionally, with similar latency, GmNet-S3 delivers a 1.7% improvement on the accuracy over MobileOne-S4. GmNet-S4 achieves 2x faster compared to RepViT-M1.5 on the GPU and it surpasses MobileOne-S4 of 2.1% under the similar latencies of both GPU and Mobile. LeViT-256 Graham et al. (2021) matches the accuracy of GmNet-S4 but runs twice as slow on a GPU and 16 times slower on an iPhone 14 The strong performance of GmNet can be largely attributed to the clear insights of gating mechanisms and simplest architectures. Fig. 6 further illustrates the latency-accuracy trade-off across different models. GmNet variants achieve substantially lower latency compared to related works, while maintaining competitive or superior Top-1 accuracy. More comparisons and results can be found in supplementary.

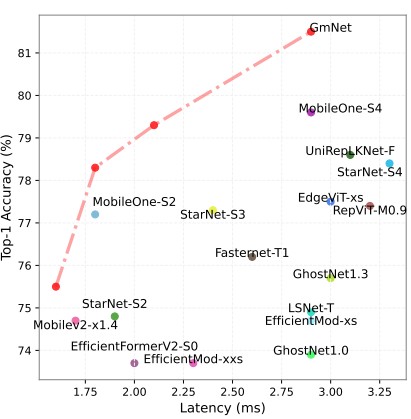

Figure 6: Trade-off between Top-1 accuracy and latency on A100.

## 5.2 ABLATION STUDIES

**More studies on different activation functions.** To further explore the effect of different activation functions, we trained various GmNet-S3 variants on ImageNet-1k. As illustrated in Fig. 5, we replaced ReLU6 with GELU, ReLU or remove the activation function. To better reflect the differences between different models, we set the radii to a larger range/

As shown in the Table. 2, we can find that, the increases on classifying the high-frequency components are significant comparing models using and not using the activation functions. For example, comparing results of 'Identity' and 'ReLU' with the improvement of 11% on the raw data, improvement on high-frequencies is over 3 times on average. 'GELU' and 'ReLU' shows advances on low-/high- frequency components respectively compared to each other. This aligns with our understanding of how different types of activation functions

Table 2: The accuracies of classifying the raw data and their low-/high-frequency components under different activation functions on ImageNet-1k. We gradually increase the radii by a step of 12. This result is the average of five testings.

| Activation | Identity | | ReLU | | GELU | | ReLU6 | |
|---|---|---|---|---|---|---|---|---|
| Raw data | 70.5 | | 78.3 | | 78.4 | | 79.3 | |
| Frequency | Low | High | Low | High | Low | High | Low | High |
| $r = 12$ | 9.79 | 12.6 | 12.0 | 45.9 | 12.7 | 41.5 | 14.8 | 51.7 |
| $r = 24$ | 38.1 | 1.7 | 38.6 | 13.5 | 40.0 | 9.4 | 41.6 | 12.1 |
| $r = 36$ | 52.9 | 0.7 | 56.2 | 4.9 | 58.7 | 3.9 | 55.2 | 4.7 |
| $r = 48$ | 63.2 | 0.5 | 64.5 | 2.3 | 66.1 | 2.1 | 64.4 | 2.5 |
| $r = 60$ | 66.6 | 0.9 | 69.4 | 1.0 | 70.7 | 1.1 | 71.1 | 1.4 |

impact frequency response. Notably, the closer performance of models with Identity and ReLU/GELU at low frequencies suggests the low-frequency bias of convolution-based networks.

Moreover, even considering the improvements on the raw data, model using the ReLU6 shows obvious increase on the high-frequency components compared to the model using GELU especially when we set $r$ to 12, 24, 36. Compared to the model with ReLU, ReLU6 is more effective in preventing overfitting to high-frequency components since it has better performance on low-frequencies. Considering performances of ReLU, GELU, and ReLU6, we can observe that achieving better performance on high frequencies at the expense of lower frequencies does not necessarily lead to overall improvement, and vice versa. To get a better performance on the raw data, it is essential to enhance the model's ability to learn various frequency signals.

**Comparison with existing methods from the frequency perspective.** As addressed in Table 2, a model should achieves strong performance across different frequency components to deliver a

Table 4: Comparison of different GLU designs for GmNet-S3 on ImageNet-1K. Here, LN, DW, and Pool represent layer normalization, depth-wise convolution with a kernel size of 3, and average pooling with a 3×3 window, respectively. We underline all notable scores in classifying the different frequency decompositions. Considering gaps of overall performances, an improvement which is remarkable should exceed 1.0. This result is the average of five testings. We also provide more variants of GLUs in the supplementary materials.

| GLUs | Top-1 (%) | Params (M) | GPU (ms) | $r=12$ | | $r=24$ | | $r=36$ | | $r=48$ | | $r=60$ | |
|---|---|---|---|---|---|---|---|---|---|---|---|---|---|
| | | | | Low | High | Low | High | Low | High | Low | High | Low | High |
| $\sigma(x) \cdot \text{LN}(x)$ | 78.9 | 7.8 | 2.9 | 12.1 | 47.6 | 41.6 | 10.9 | 56.4 | 5.2 | 64.7 | 2.4 | 69.8 | 1.2 |
| $\sigma(x) \cdot \text{DW}(x)$ | 79.0 | 8.0 | 2.4 | 12.3 | 49.0 | 42.7 | 9.6 | 58.1 | 4.6 | 65.7 | 2.3 | 71.2 | 1.1 |
| $\sigma(x) \cdot (x - \text{Pool}(x))$ | 78.6 | 7.8 | 2.4 | 14.2 | 50.1 | 42.3 | 10.8 | 55.8 | 4.9 | 63.8 | 2.7 | 69.9 | 1.3 |
| $\sigma(x) \cdot \text{FC}(x)$ | 79.2 | 20.2 | 3.6 | 10.8 | 51.4 | 39.6 | 8.7 | 52.6 | 4.4 | 62.9 | 3.4 | 69.9 | 2.4 |
| $\sigma(x) \cdot x$ | 79.3 | 7.8 | 2.1 | 14.8 | 51.7 | 41.6 | 12.1 | 55.2 | 4.7 | 64.4 | 2.5 | 71.1 | 1.4 |

better overall performance. However, both pure convolutional architectures and transformers exhibit a low-frequency bias, as discussed in Bai et al. (2022); Tang et al. (2022). Therefore, enhancing the performance of a lightweight model depends on its ability to more effectively capture high-frequency information. To address the advantages of GmNet on overcoming the low-frequency bias, we test some existing models on different frequency components of different radii. We select three kinds of typical lightweight methods for comparison including pure conv-based model MobileOne-S2 Vasu et al. (2023b), attention-based model EfficientMod-xs Ma et al. (2024b) and model also employing GLUs-like structure StarNet-S4 Ma et al. (2024a). As shown in Table 3, accuracies of low-frequency components are close

Table 3: Comparison with recent methods. We test models on the high-/low-frequency components on the ImageNet-1k. The highest values of each columns are highlighted.

| Methods | Top-1 (%) | $r=12$ | | $r=24$ | | $r=36$ | |
|---|---|---|---|---|---|---|---|
| | | High | Low | High | Low | High | Low |
| MobileOne-S2 Vasu et al. (2023b) | 77.4 | 35.0 | 11.6 | 6.5 | 36.9 | 2.4 | 53.5 |
| EfficientMod-xs Ma et al. (2024b) | 78.3 | 45.4 | 12.9 | 9.4 | 40.6 | 3.5 | 54.6 |
| StarNet-S4 Ma et al. (2024a) | 78.4 | 43.3 | 13.8 | 9.4 | 41.3 | 3.4 | 54.8 |
| GmNet-S3 | **79.3** | **51.7** | **14.8** | **12.1** | **41.6** | **4.7** | **55.2** |

among different models considering the overall performance. However, it shows that GmNet-S3 clearly surpass the other models in high frequency components. For example, GmNet-S3 has a 6.3% improvement compared to EfficientMod-xs when $r = 12$ and 2.7% increase when $r = 24$. For StarNet, which also uses a GLU-like structure with dual-channel FC, it struggles to effectively emphasize high-frequency signals. The simplest GLUs design can achieve a better balance between the efficiency and the effectiveness.

**Study on designs of the GLU.** In GmNet, the gated linear unit adopts the simplest design, which can be defined as $\sigma(x) \cdot x$. For comparison, we modify the GLU design and conduct experiments to test performance on raw data as well as on decompositions at different frequency levels. As shown in the Table 4, the simplest design achieve the best performance both on effectiveness and efficiency for the overall performance. For the decomposed frequency components, we observe clear differences among various GLU designs. The GLU of $\sigma(x) \cdot x$ demonstrates significantly higher accuracy in classifying high-frequency components. For example, for $r = 12$ and $r = 24$, the GLU with $\sigma(x) \cdot x$ shows an improvement of 4.1 over the LN design and 2.5 over the DW design. This indicates that the simplest GLU design is already effective at introducing reliable high-frequency components to enhance the model's ability to learn them. Designs aimed at smoothing information show a notable improvement in some low-frequency components. For instance, with similar overall performance, the GLUs using $\sigma(x) \cdot \text{DW}(x)$ and $\sigma(x) \cdot \text{LN}(x)$ achieve better results on low-frequency components when the radii are set to 24, 36, and 48. The model using a linear layer in GLUs offers performance comparable to GmNet-S3 and is adept at learning low-frequency features. However, its placement at a high-dimensional stage is problematic. This design choice leads to an excessive number of parameters and a significant increase in latency. Moreover, depth-wise convolution is more effective than layer normalization in encouraging neural networks to learn from low-frequency components which is also more efficient. For the design with the average pooling, it does not perform better in classifying high-frequency signals. This may be because $x - \text{pool}(x)$ acts as an overly aggressive high-pass filter, which does not retain the original high-frequency signals in $x$ well and instead introduces more high-frequency noise.

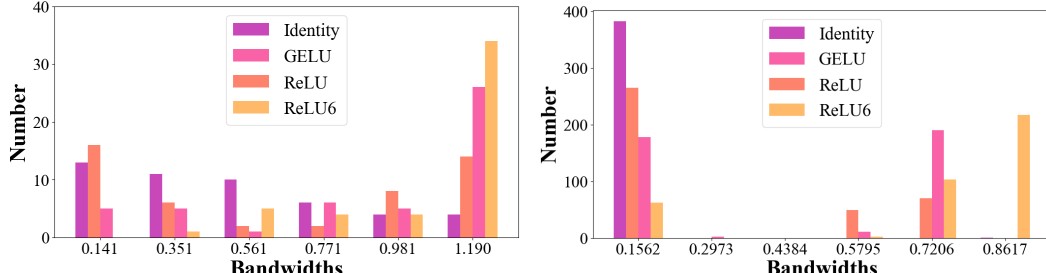

Figure 7: The histogram illustrates the distribution of bandwidths of convolution kernels. Bandwidths represents the capability of a convolution kernel for capturing various frequency information. We use weights of the convolution layer which under the GLU in the first block (left) and the last block (right) of the GmNet-S3. All modals are trained on the raw data of the ImageNet-1k. In general, the further the distribution shifts to the right, the stronger the convolutional kernel's ability to capture signals of different frequencies.

**Bandwidths analysis of convolution kernels.** As discussed in the Tang et al. (2022), the convolution layer may play roles of 'smoothing' the feature which means it has a low-frequency bias. Experiments on studying weights of the convolution layer is insightful to give more evidences of how GLUs effect the learning of different frequency components Wang et al. (2020); Tang et al. (2022); Bai et al. (2022). In this paper, we propose using the bandwidths of convolution kernels to represent their ability of responding to different frequency components. Specifically, a wider bandwidth indicates that the kernel can process a broader range of frequencies, allowing it to capture diverse frequency components simultaneously and thereby preserve rich information from the feature. As illustrated in Figure 7, the distributions of the ReLU model suggest that its convolution kernels tend to focus on a narrow range of frequency components leading to relatively lower bandwidths. It indirectly reflects an overemphasis on high-frequency components. Although the model using GELU exhibits a better distribution in the top convolutional layers, it still has a low-frequency bias, leading to a distribution shift in the bottom convolutional layers. Compared to other activations, the enhanced bandwidth distribution of the model using ReLU6 demonstrates better generalization for this task. The properties of the convolution kernels align with results in Table 2.

**Contribution breakdown.** We show the ablation study of different block designs and adjust the dimensions/number of blocks under strictly matched FLOPs, parameter count, and training settings, and report the mean ± std over three random seeds. We replace the 7×7 dwconv with a linear layer and replace the GLU with a ReLU as the baseline block. The results are shown in Table 5. Overall, Table demonstrates that the performance gains introduced by gating mechanism are distinct improvements beyond what other components alone can offer.

Table 5: Contribution breakdown under matched FLOPs and parameters (mean ± std).

| Variant | Params (M) | FLOPs (G) | Top-1 Acc (%) |
|---|---|---|---|
| Baseline | 7.82 | 1.24 | 71.5 ± 0.2 |
| + 7×7 DWConv | 7.82 | 1.28 | 78.1 ± 0.1 |
| + Gate (Identity) | 7.82 | 1.24 | 69.2 ± 0.3 |
| + Gate (ReLU) | 7.82 | 1.24 | 78.0 ± 0.2 |
| + Gate (GELU) | 7.82 | 1.24 | 77.9 ± 0.1 |
| + Gate (ReLU6) | 7.82 | 1.24 | 78.5 ± 0.1 |
| + ReLU6 | 7.82 | 1.24 | 77.9 ± 0.1 |
| Full GmNet | 7.82 | 1.24 | **79.2 ± 0.1** |

## 6  CONCLUSION

This paper tackled the prevalent low-frequency bias in lightweight networks through a novel frequency-based analysis of gating mechanisms. We found that in a Gated Linear Unit (GLU), element-wise multiplication introduces valuable high-frequency information, while the paired activation function provides crucial control to filter for useful signals over noise. Our resulting model, the Gating Mechanism Network (GmNet), validates this approach by setting a new state-of-the-art in efficient network design. This work demonstrates that a frequency-aware methodology is a promising path toward creating future models that are both efficient and representationally robust.

# 7 STATEMENTS

## 7.1 ETHICS STATEMENT

In our paper, we strictly follow the ICLR ethical research standards and laws. To the best of our knowledge, our work abides by the General Ethical Principles.

## 7.2 REPRODUCIBILITY STATEMENT

We adhere to ICLR reproducibility standards and ensure the reproducibility of our work. All datasets we employed are publicly available. We will provide the code to reviewers and area chairs in the supplementary material.

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

# A  APPENDIX

## A.1  IMPLEMENTATION DETAILS

### A.1.1  PSEUDO-CODES OF MODEL ARCHITECTURES

In our modified ResNet18, featured in Fig. 2, we adjust the activation function as the different variants. As an example, we provide the pseudo-codes of Res18-Gate-ReLU in the Algorithm 1

---
**Algorithm 1** Pseudo-codes of Res18-Gate-ReLU
---
```
def Block(x, in_planes, planes)
    out = Conv2d(x, in_planes, planes, 3, 1, 1)
    out = BatchNorm2d(x)
    out = ReLU(out) * out
    out = Conv2d(out, planes, planes, 3, 1, 1)
    out = BatchNorm2d(out)
    out += self.shortcut(x)
    out = ReLU(out)
    return out
```
---

For the proposed GmNet, featured in Fig. 4, we provide the pseudo-code of GmNet in the Algorithm 2. Also, for ease of reproduction, we include a separate file in the supplementary materials dedicated to our GmNet.

---
**Algorithm 2** Pseudo-codes of GmNet
---
```
def Block(x, dim, mlp_ratio)
    input = x
    x = DWConv2d(x, dim, dim, 7, 1, 3, group=dim)
    x = BatchNorm2d(x)
    x = Conv2d(x, dim, mlp_ratio*dim, 1)
    x = ReLU6(x) * x
    x = Conv2d(x, mlp_ratio*dim, dim, 1)
    x = BatchNorm2d(x)
    x = DWConv2d(x, dim, dim, 7, 1, 3, group=dim)
    x = input + drop_path(layer_scale(x))
    return x
```
---

### A.1.2  TRAINING RECIPES

We first provide the detailed training settings of variants of ResNet18 in Table 7.

### A.1.3  GMNET VARIANTS

We provide the setting of variants of GmNet in Table. 6.

| Variant | $C_1$ | depth | ratio | Params | FLOPs |
|---|---|---|---|---|---|
| GmNet-S1 | 40 | $[2, 2, 10, 2]$ | $[3, 3, 3, 2]$ | 3.7 M | 0.6 G |
| GmNet-S2 | 48 | $[2, 2, 8, 3]$ | $[3, 3, 3, 2]$ | 6.2 M | 0.9 G |
| GmNet-S3 | 48 | $[3, 3, 8, 3]$ | $[4, 4, 4, 4]$ | 7.8 M | 1.2 G |
| GmNet-S4 | 68 | $[3, 3, 11, 3]$ | $[4, 4, 4, 4]$ | 17.0 M | 2.7 G |

Table 6: Configurations of GmNet. We vary the embedding width, depth, and gating ratio to construct different model sizes of GmNet.

We also provide a detailed training configures of GmNet in this section as shown in the Table 8.

| config | value |
|---|---|
| image size | 32 |
| optimizer | SGD |
| base learning rate | 0.1 |
| weight decay | 5e-4 |
| optimizer momentum | 0.9 |
| batch size | 128 |
| learning rate schedule | cosine decay |
| training epochs | 100 |

Table 7: Res18 variants training setting

| config | value |
|---|---|
| image size | 224 |
| optimizer | AdamW |
| base learning rate | $3e-3$ |
| weight decay | 0.03 |
| optimizer momentum | $\beta_1, \beta_2 = 0.9, 0.999$ |
| batch size | 2048 |
| learning rate schedule | cosine decay |
| warmup epochs | 5 |
| training epochs | 300 |
| AutoAugment | rand-m1-mstd0.5-inc1 |
| label smoothing | 0.1 |
| cutmix | 0.4 |
| color jitter | 0. |
| drop path | 0.(S1/S2), 0.02(S3/ S4) |

Table 8: GmNet training setting

## A.2 MORE GLUS DESIGNS.

In this section, we present additional block designs to demonstrate the efficiency and effectiveness of our proposed architecture. As shown in Table 9, we conducted experiments incorporating fully connected (FC) layers into the Gated Linear Units (GLUs). Adding an extra FC layer to one branch of the GLUs may slightly improve performance. However, this modification significantly increases the number of parameters and reduces the model's latency.

Moreover, this enhanced architecture does not perform better in classifying different frequency components. The reason is that features processed by an FC layer cannot effectively emphasize various frequency components. While adding more parameters and employing different training methods might enhance the capability to learn different frequency components, in lightweight models, the simplest GLU design often delivers better performance. This observation is consistent with findings from many recent studies on lightweight models.

## A.3 MORE COMPARISONS OF THE MAIN RESULTS.

We provide more comparisons of the main results in Fig. 8. We plot a larger latency-acc trade-off figure to include more methods including EdgeViT, MobileNetV2 and GhostNet.

## A.4 RESULTS ON CUB-100.

We evaluated GmNet-S1 on the CUB-100 dataset. The results are competitive, as shown in Table 10. Compared to ShuffleNet-V2, GmNet-S1 achieves better performance with a smaller model size.

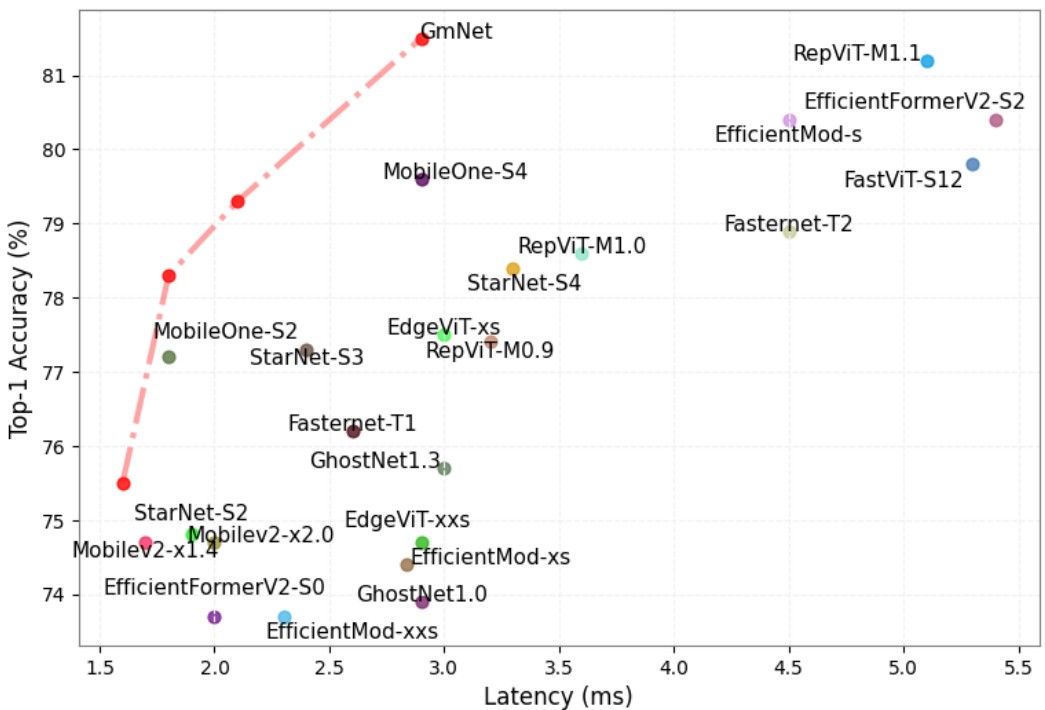

Figure 8: Top-1 accuracy vs Latency on A100. Our models have significantly smaller latency compared to related works.

Table 9: Comparison of different GLU designs for GmNet-S3 on ImageNet-1K.

| GLUs | Top-1 | Params | GPU | $r = 12$ | | $r = 24$ | | $r = 36$ | | $r = 48$ | | $r = 60$ | |
|---|---|---|---|---|---|---|---|---|---|---|---|---|---|
| | (%) | (M) | (ms) | Low | High | Low | High | Low | High | Low | High | Low | High |
| $\sigma(x) \cdot \text{FC}(x)$ | 79.2 | 20.2 | 3.6 | 10.8 | 51.4 | 39.6 | 8.7 | 52.6 | 4.4 | 62.9 | 3.4 | 69.9 | 2.4 |
| $\sigma(\text{FC}(x)) \cdot x$ | 79.6 | 20.2 | 3.4 | 9.4 | 48.9 | 35.0 | 9.1 | 51.1 | 3.6 | 62.1 | 3.1 | 68.7 | 2.4 |
| $\sigma(x) \cdot x$ | 79.3 | 7.8 | 2.1 | 14.8 | 51.7 | 41.6 | 12.1 | 55.2 | 4.7 | 64.4 | 2.5 | 71.1 | 1.4 |

## A.5 Performance with SwiGLU and SiLU

We have conducted experiments with the SiLU variant on CIFAR-10 to illustrate the learning dynamics in Fig. 14. Following the reviewer's advice, we adapted both SwiGLU and SiLU to GmNet-S3 and trained the models under the same settings. The results in Table 11 show that both activations improve performance compared to the baseline configuration, which removes activations inside the GLU. However, their gains remain consistently smaller than those achieved by our proposed design, confirming that the proposed GmNet block is better aligned with the frequency characteristics and architectural constraints of lightweight models.

## A.6 Performance of changing ReLU6 to ReLU.

Table 12 shows the performance change when replacing ReLU6 with ReLU across multiple GmNet variants. The degradation is consistent across all model sizes.

## A.7 Studies on aliasing and robustness

We train **Res18-Gate-ReLU** and **Res18-Gate-GELU** on CIFAR-10 following the setting of **?** to show how different activation functions affect robustness. We conducted an adversarial robustness ablation under PGD attacks. As shown below, the ReLU-based GLU—which emphasizes higher-frequency components—exhibits slightly lower PGD accuracy:

| Variant | Params (M) | Top-1 Acc (%) |
|---|---|---|
| ShuffleNet-V2 | 3.5 | 76.0 |
| GmNet-S1 | 3.1 | 81.5 |

Table 10: Results on CUB-100.

| Model / Activation | Top-1 Acc (%) |
|---|---|
| Baseline (Identity) | 70.5 |
| + SiLU | 77.9 |
| + SwiGLU | 77.2 |
| + Proposed (ReLU6-GLU) | **79.3** |

Table 11: Results of using SiLU and SwiGLU in GmNet-S3.

- **Res18-Gate-ReLU:** 45.7% PGD accuracy
- **Res18-Gate-GELU:** 46.7% PGD accuracy

Clean accuracy remains similar for both variants (82.2% vs. 82.8%), but the ∼1% drop in PGD robustness for the ReLU gate is consistent with prior findings that stronger high-frequency reliance increases vulnerability to aliasing and adversarial perturbations.

### A.8 EXPERIMENTS ON CONVNEXT

To show the generalization of gating structure, we adapted our GLU to ConvNeXt-Tiny and trained the model using the same setting as GmNet on ImageNet-1k. As shown in Table 13, GLU improves the performance without introducing any additional computational cost.

The improvement is smaller compared to GmNet, which may result from the fact that large CNNs are less affected by activation-induced frequency bias. Block design should be architecture-dependent.

### A.9 MORE RESULTS ON EFFORMERV2 AND MOBILENETV2.

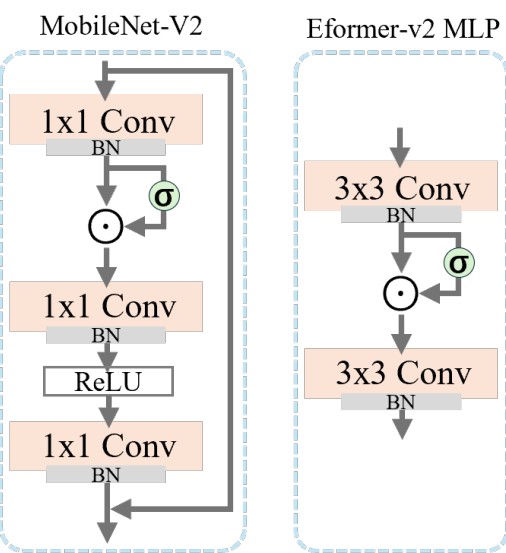

Figure 9: The illustration of modifications of MobileNetV2 and EfficientFormer-V2.

We first show the illustration of the modifications for both models in . Moreover, we plot the testing curves with different settings of thresholds. As shown in Fig. 13, the overall performance trend is consistent with the charts and figures presented in the main text. The model using ReLU6 shows

| Variant | ReLU6 $\rightarrow$ ReLU |
|---|---|
| GmNet-S1 | $75.5 \rightarrow 74.7$ |
| GmNet-S2 | $78.3 \rightarrow 77.4$ |
| GmNet-S3 | $79.3 \rightarrow 78.3$ |
| GmNet-S4 | $81.5 \rightarrow 80.5$ |

Table 12: Performance degradation when replacing ReLU6 with ReLU.

| Model | Top-1 Acc (%) | Params (M) | FLOPs (G) |
|---|---|---|---|
| ConvNeXt | 82.5 | 28.6 | 4.46 |
| ConvNeXt + GLU | **83.0** | 28.6 | 4.46 |

Table 13: Comparison between ConvNeXt-Tiny and its GLU-enhanced variant.

better performance on the high-frequency components where the overall performance also surpasses other models. Meanwhile, the model using GELU performs better on low-frequencies. The extra results demonstrate the effect of GLU on helping model learning various frequency information and the improvements on high-frequency information have more impact on the final performance.

## A.10 QUANTITATIVE SPECTRAL EVIDENCES.

To further explore the effect of different activation functions, we provide the quantitative spectral evidence. Based on the GmNet-S3 variants, we computed high/low-frequency energy ratios across multiple layers and model variants as shown in Table 14. Firstly, we extract the layers before/after the gate and compute their high/low-frequency energy ratios to show the spectral changes. Also, to demonstrate how different activation functions affect the model's frequency response, we compute the high/low-frequency energy ratios of the first DW-Conv layers in each stages.

We define the low frequencies as the central 1/4 region of the 2D spectrum. Table 14 shows that a consistent spectral pattern distinguishing smooth and non-smooth activations. For GELU, the transition from f to g typically increases the low-frequency response, and across all stages GELU yields the lowest high/low ratios. This aligns with its smooth functional form, which naturally biases the network toward low-frequency representations. In contrast, ReLU and ReLU6 systematically amplify high-frequency components. In the early stages (Stage 0 and 1), ReLU exhibits the strongest high-frequency response, reflecting its non-smooth activation behavior and its tendency to preserve or enhance sharp transitions. In deeper layers (Stage 2 and 3), ReLU6 produces the highest high/low ratios, suggesting that its clipped nonlinearity becomes more influential as depth increases—potentially explaining why the ReLU6-based model obtains the best overall performance. These effects are stable across stages, blocks, and models, and they directly support our hypothesis: smooth activations such as GELU favor low-frequency features, whereas non-smooth activations (ReLU/ReLU6) amplify high-frequency content. This yields a clear falsifiable prediction—if a smooth activation were ever to systematically exceed ReLU/ReLU6 in high-frequency ratios under comparable settings, our hypothesis would be invalidated—which strengthens the explanatory robustness of the spectral analysis presented in Sec. 3.2.

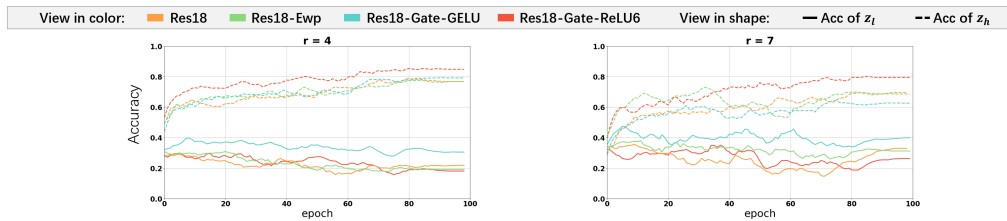

Figure 10: Additional results under different threshold configurations of different variants of ResNet18.

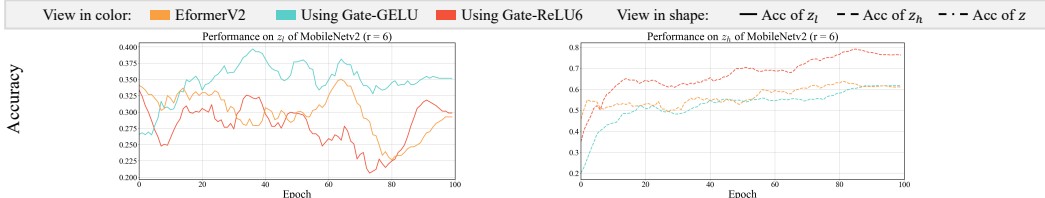

Figure 11: Additional results under different threshold configurations of different variants of MobileNetv2.

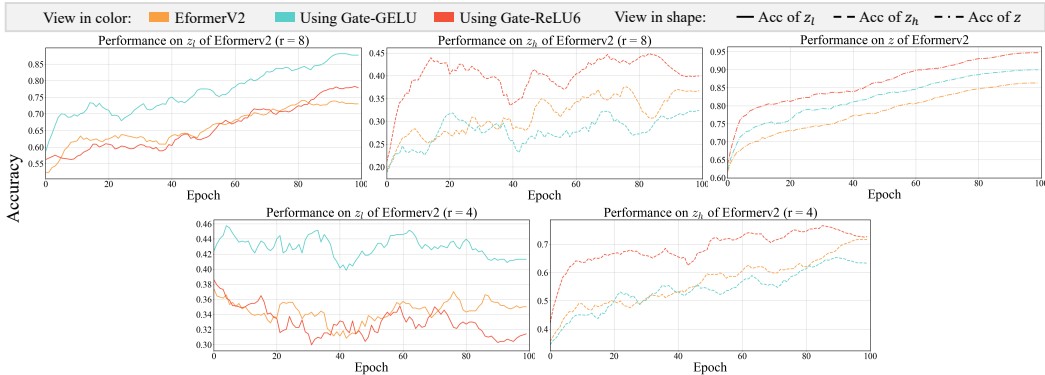

Figure 12: Additional results under different threshold configurations of different variants of Efficientformer-V2.

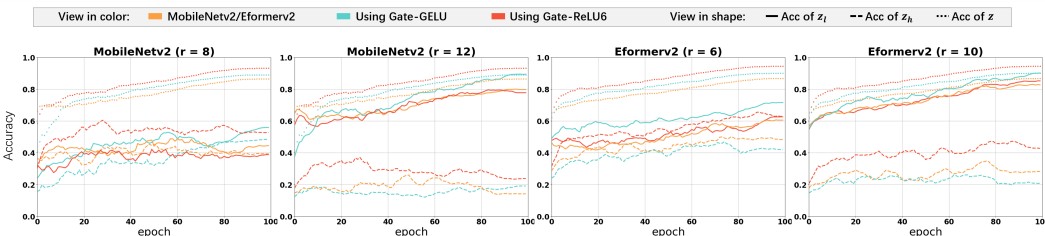

Figure 13: Additional results under different threshold configurations of different variants of MobileNetv2 and EfficientFormer-v2 (Eformerv2).

Table 14: High/Low Frequency Ratio Comparison. We computed the spectral changes from the layer before to the layer after the gate which are defined as f and g respectively. We also compute the high/low-frequency energy ratios of the first 7×7 DW-Conv layers of each stage.

| High/Low Frequency Ratio Changes before/after the gate variants | | | | |
|---|---|---|---|---|
| Stage | Layer Pair | ReLU6 (H/L) | GELU (H/L) | ReLU (H/L) |
| 0.1 | $f \to g$ | $0.1195 \to 0.1200$ | $0.0575 \to 0.0040$ | $0.5172 \to 0.5674$ |
| 1.1 | $f \to g$ | $0.0989 \to 0.1429$ | $0.0423 \to 0.0022$ | $0.1422 \to 0.1751$ |
| 2.1 | $f \to g$ | $0.0386 \to 0.0706$ | $0.0252 \to 0.0013$ | $0.0018 \to 0.0030$ |
| 3.1 | $f \to g$ | $0.0019 \to 0.0281$ | $0.0106 \to 0.0003$ | $0.0032 \to 0.0341$ |
| High/Low Frequency Ratio Comparison at DW-Conv layers | | | | |
| Stage | Layer | GELU (H/L) | ReLU (H/L) | ReLU6 (H/L) |
| 0.1 | 1st DW-Conv | 0.1203 | **1.1553** | 0.7057 |
| 1.1 | 1st DW-Conv | 0.0695 | **0.3040** | 0.2452 |
| 2.1 | 1st DW-Conv | 0.0381 | 0.0024 | **0.0697** |
| 3.1 | 1st DW-Conv | 0.0057 | 0.0088 | **0.0142** |

## A.11 EXTRA EXPERIMENTS ON GATING MECHANISMS.

In this section, we present additional experimental results demonstrating how different activation functions affect frequency learning. Firstly, we have conducted experiments with different settings of $r$ of Fig. 3 in the main paper. We have set $r$ to $\{4, 7\}$ respectively. The results are shown in Fig., the performances are aligned with our analysis in Sec. 3 which indicates that the results shown in Fig. 3 are not accidental.

Specifically, we conduct experimearents using another smooth activation function, Swish (SiLU), and a non-smooth function, ReLU6.

As shown in Figures 14a and 14b, ReLU6 performs similarly to GELU. Although ReLU6 may introduce more high-frequency components, it does not perform as well as ReLU for two main reasons: (1) ReLU6 caps the activation values. It can reduce the model's sensitivity to high-frequency components where those components are often associated with higher activation values. (2) The use of low-resolution images can adversely affect the performance in classifying high-frequency components, as finer details are lost, making it harder for the model to learn these features. Figures 14c and 14d present comparisons between SiLU (Swish) and ReLU, as well as between SiLU and GELU, respectively. The Res18-Gate-SiLU performs better on lower-frequency components, specifically in the range $r \in (0, r_1)$. This indicates that SiLU has a greater smoothing effect on the information, encouraging the model to learn more effectively from lower frequencies.

Moreover, we investigated the impact of different training strategies to understand their effects on model performance. Specifically, as plotted in the Fig. 15, we replaced the optimizer with Adam, setting its learning rate to 0.001. While the training curves showed noticeable variations compared to the baseline setup, the overall performance differences among the model variants remained consistent. This consistency indicates that the observed behaviors are robust to changes in optimization strategies.

We also conduct experiments on setting the redii to different sets. As shown in Figs. 16 and 17, we set radii to $[0, 4, 8, 12, +\infty]$ and $[0, 8, 16, 24, +\infty]$ respectively. When the frequency intervals are too small, the differences between the methods become less pronounced, especially in the lower-frequency components. When the frequency intervals are too large, all models struggle to classify higher-frequency components. Although differences between models become more pronounced in the lower-frequency components, such as between GELU and ReLU activations, to better understand the training dynamics, it is necessary to examine the differences in the high-frequency components as well. Therefore, we decide to display the results of setting radii to $[0, 6, 12, 24, +\infty]$ in the main body of the paper to have a better understand of the training dynamic of different variants. These results provide valuable insights into the functionality of the gating mechanisms. They suggest that the interaction between the element-wise product and the activation functions is a general phenomenon.

## A.12 DOWNSTREAM TASKS.

We further provide the results of downstream tasks of *Object Detection*, *Instance Segmentation* and *Semantic Segmentation*. Firstly, we conducted experiments on GmNet-S3 on MSCOCO 2017 with the Mask RCNN

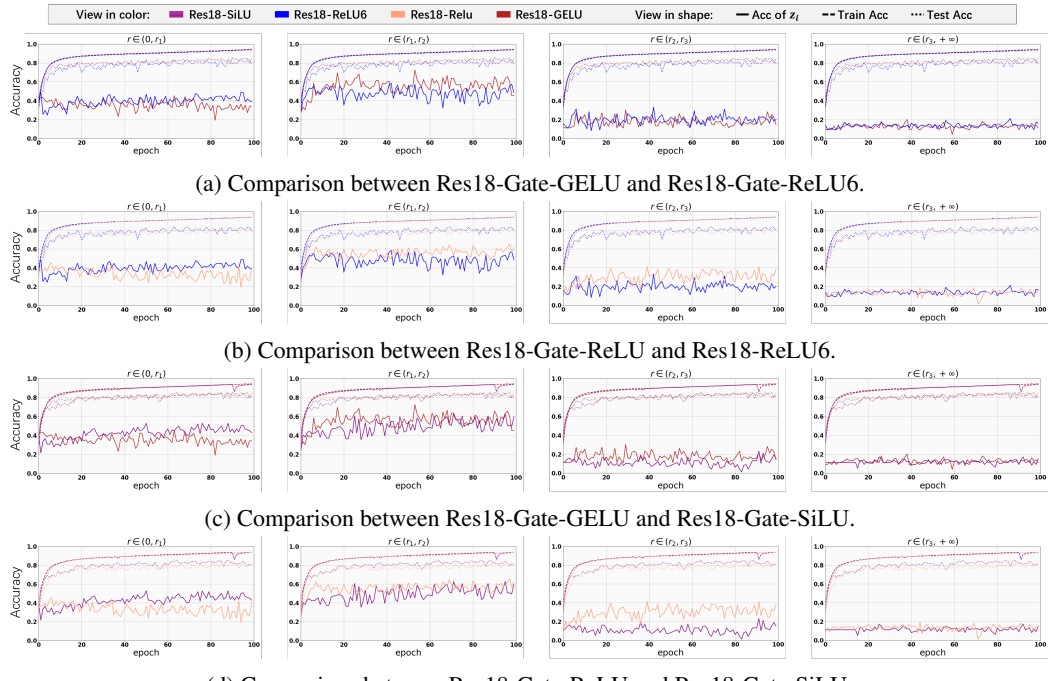

(a) Comparison between Res18-Gate-GELU and Res18-Gate-ReLU6.

(b) Comparison between Res18-Gate-ReLU and Res18-ReLU6.

(c) Comparison between Res18-Gate-GELU and Res18-Gate-SiLU.

(d) Comparison between Res18-Gate-ReLU and Res18-Gate-SiLU.

Figure 14: Learning curves of Resnet18 and its variants for 100 epoch. The radii are set to $\{0, 6, 12, 18, +\infty\}$

framework for object detection and instance segmentation. Our method shows better performance compared to the existing methods RepViT-M1.5 Wang et al. (2024) and

Table 15: Object detection & Instance segmentation& Semantic segmentation. The latency is tested on iPhone 14 by Core ML Tools.

| Backbone | Latency↓ (ms) | Object Detection↑ | | | Instance Segmentation↑ | | | Semantic ↑ mIoU |
|---|---|---|---|---|---|---|---|---|
| | | $AP^{box}$ | $AP^{box}_{50}$ | $AP^{box}_{75}$ | $AP^{mask}$ | $AP^{mask}_{50}$ | $AP^{mask}_{75}$ | |
| EfficientFormer-L3 | 12.4 | 41.4 | 63.9 | 44.7 | 38.1 | 61.0 | 40.4 | 43.5 |
| RepViT-M1.5 | 6.9 | 41.6 | 63.2 | 45.3 | 38.9 | 60.5 | 41.5 | 43.6 |
| GmNet-S3 | 5.2 | 42.2 | 63.4 | 46.7 | 40.1 | 61.2 | 42.9 | 44.6 |

EfficientFormer-L3 Li et al. (2022) with better efficiency in terms of latency, $AP^{box}$ and $AP^{mask}$ under similar model sizes. Specifically, GmNet-S3 outperforms RepViT-M1.5 significantly by 2.4 $AP^{mask}_{75}$ and 1.4 $AP^{box}_{75}$. Meanwhile, GmNet-S3 has 1.7 ms faster on the Mobile latency and more than 2 times faster than EfficientFormer-L3. For the semantic segmentation, we conduct experiments on ADE20K to verify the performance of GmNet-S3. Following the existing methods, we integrate GmNet into the Semantic FPN framework. With significant improvements on the speed, GmNet-S3 still match the performance on semantic segmentation task with RepV-T-M1.5 and EfficientFormer-L3.

## A.13 THE USAGE OF LARGE LANGUAGE MODELS (LLMS)

We used GPT for polishing grammar and improving readability. All research ideas and analyses were conducted by the authors, who take full responsibility for the content.

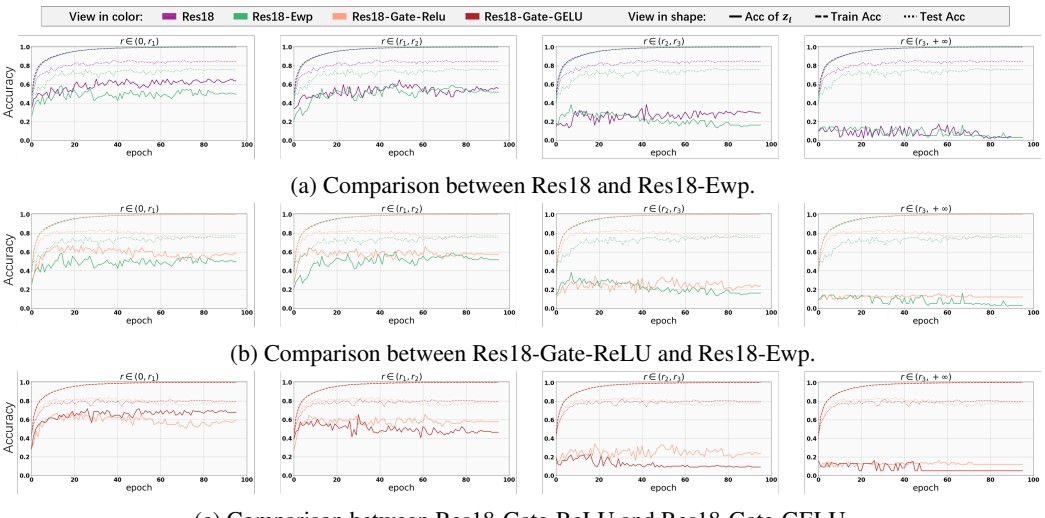

(a) Comparison between Res18 and Res18-Ewp.

(b) Comparison between Res18-Gate-ReLU and Res18-Ewp.

(c) Comparison between Res18-Gate-ReLU and Res18-Gate-GELU.

Figure 15: Learning curves of Resnet18 and its variants for $100$ epochs with optimizer of Adam. The learning rate is set to $0.001$. Radii are set to $\{0, 6, 12, 18, +\infty\}$

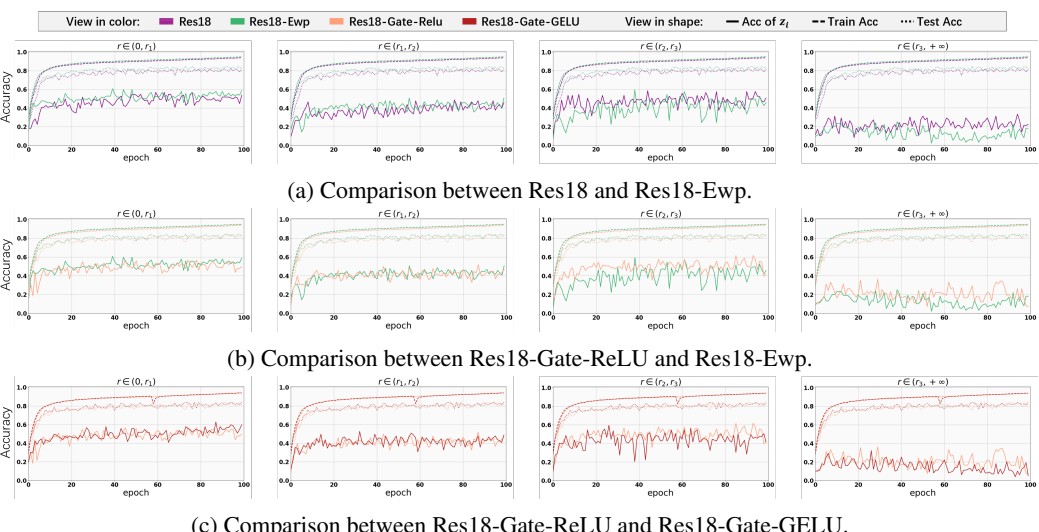

(a) Comparison between Res18 and Res18-Ewp.

(b) Comparison between Res18-Gate-ReLU and Res18-Ewp.

(c) Comparison between Res18-Gate-ReLU and Res18-Gate-GELU.

Figure 16: Learning curves of Resnet18 and its variants for $100$ epochs with optimizer of SGD. Radii are set to $\{0, 4, 8, 12, +\infty\}$. When the frequency intervals are too small, the differences between the methods become less pronounced, especially in the lower-frequency components. However, it remains evident that the different variants have distinct effects on learning the various frequency components.

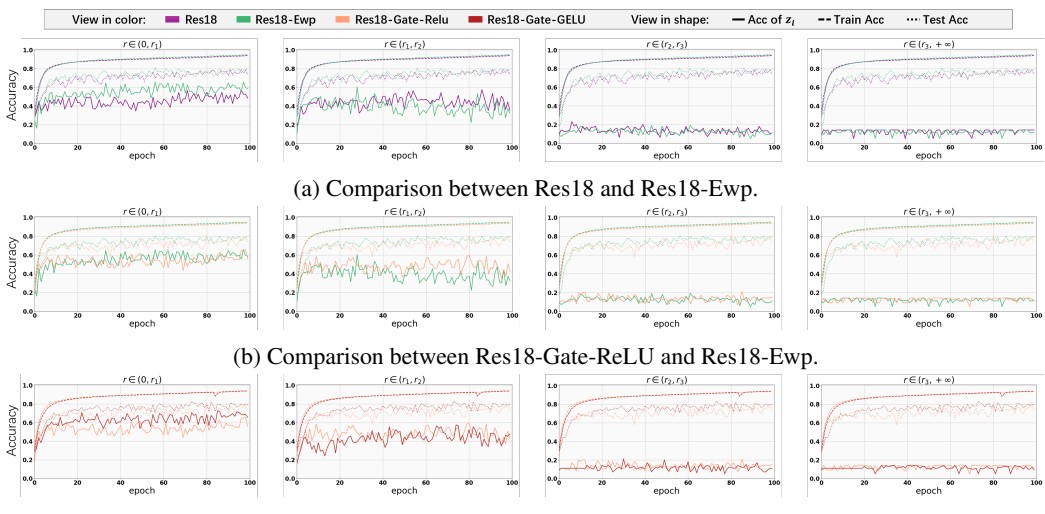

(a) Comparison between Res18 and Res18-Ewp.

(b) Comparison between Res18-Gate-ReLU and Res18-Ewp.

(c) Comparison between Res18-Gate-ReLU and Res18-Gate-GELU.

Figure 17: Learning curves of Resnet18 and its variants for 100 epochs with optimizer of SGD. Radii are set to $\{0, 8, 16, 24, +\infty\}$. When the frequency intervals are too large, all models struggle to classify higher-frequency components. Although differences between models become more pronounced in the lower-frequency components, such as between GELU and ReLU activations, to better understand the training dynamics, it is necessary to examine the differences in the high-frequency components as well.

