# OpenReview forum: "GmNet: Revisiting Gating Mechanisms From A Frequency View"
_ICLR.cc/2026/Conference — ICLR 2026 Poster_

### Official Review · Reviewer_KXHh · 2025-10-27

**Soundness:** 3
**Presentation:** 2
**Contribution:** 2
**Rating:** 6
**Confidence:** 2

**Summary:**

The paper deals with lightweight networks for edge devices. The propose to use gating mechanisms like GLUs to directly counteract the low-frequency bias present in many efficient architectures. They analyse GLUs from a frequency perspective and present a lightweight architecture they call Gating Mechanism Network (GmNet) on top of MobileNetV2. The study different gating mechanisms and show that the simplest linear one performs best. They get state-of-the-art resuts on ImageNet

**Strengths:**

* The paper analyses GLUs from a frequency perspective and links their core operations to the ability to modulate a network’s spectral response. They use that to counteract a low-frequency bias in lightweioght architectures
* It is nice to see that linear gating seems to perform best
* The paper reports strong results on ImageNet1k when basing GmNet on MobileNetV2 (2018). It also gets strong results on COCO for Object detection & segmentation.

**Weaknesses:**

1) The paper offers limited theoretical depth and bases the analysis on empirical Fourier observations. The analysis section (Sec 3) is weak in scope and effectively only studies how a smooth (Gelu) and "non-smooth" (ReLU6) activation function affect classification performance of a MobileNet durign training for low/high frequency components of the input image.

2) There is a lot of emphasis on results with low radii $r$. Although it is clear that the proposed method improves over other methods for high frequencies (Table 3), in terms of overall performance, the gains are small over the state-of-the-art (<1%). It is unclear to me how top-1 overall gain relates to the analysis in Sec 3 or results on low/high frequencies.

3) The paper would be strong if results on more datasets/tasks were presented. (E.g. Places or fine-grained classification datasets like CUB/Cars etc)

**Questions:**

Q1) Which dataset do Fig3/4 refer to? Is it all on ImageNet1k?

Q2) Why is the performance across epochs important in Fig3/4? what does the x-axis tell us over different epochs (eg vs re porting final performance)? It is hard for me to understand why the authors claim that "the convergence is faster" in the discussion, when the rightmost figure shows more or less equal convergence trends.

Q3) Could GmNet be adapted to recent transformer based architectures like RepVit or Lsnet?

Q4) "As shown in Fig. 4, we present the testing accuracy curves under varying frequency thresholds" - isn't the treshold (r) set to 10 everywhere? Apologies if i misunderstood.

Q5) What is the top-1 ImageNet1k performance on GMNet-S1/2/3/4 when simply changing the ReLu6 to simple ReLU (keeping the exact same architecture otherwise)?

Q6) How do you justify that the simplest, linear gating mechanism works best (eg in Tab 4/8)?

Q7) Is there a  typo in the caption of Fig 1 where it mentions " retains the general shape of the frog"?

---

> ### Author Response · Authors · 2025-11-22
>
> We sincerely appreciate the reviewer's recognition of our work. Your thoughtful feedback has helped us improve the clarity and rigor of the work. Here is our response:
>
> ---
> ## **1. Limited anaylsis**
> We appreciate the reviewer’s feedback regarding the theoretical depth of Sec. 3. Our goal in this section was to provide a lightweight theoretical intuition—rather than a full theoretical framework—explaining why smooth and non-smooth activations behave differently in terms of frequency response.
>
> To strengthen the generality of our observations, we have conducted additional experiments beyond GeLU, ReLU6 and MobileNet in the supplementary, including
> - (i) more activation functions (e.g., SiLU, ReLU) and
> - (ii) integration into a transformer-based model EfficientFormer.
>
> These extended results show consistent trends across architectures and activation families, reinforcing that the frequency behavior we report is not specific to a single model but reflects a broader empirical phenomenon. A deeper theoretical treatment is indeed valuable but is beyond the scope of this work.
>
> ---
> ## **2. Unclear relationship between gains and analysis**
> While the absolute Top-1 gain is modest (<1%), this improvement is meaningful for lightweight models operating under strict FLOP and latency constraints. In our 1.2–1.3 GFLOP setting, a 0.7–0.9% Top-1 increase corresponds to a ~8–12% relative error reduction, and importantly, it is achieved without adding computational cost. In fact, the GLU-based block reduces inference latency in our benchmark, making the overall improvement non-trivial when jointly considering accuracy and speed. For example, compared to RepViT-M1.0, GmNet-S3 improves 0.7% with 40% faster on GPU latency.
>
> Beyond final accuracy, our analysis provides multiple lines of evidence showing that the observed gains are consistent with the frequency behavior discussed in Sec. 3 which is supported by Tables 2&3 of the paper. Since lightweight CNNs primarily exploit low-/mid-frequency information, high-frequency cues contribute only a smaller—but complementary—portion of the final decision. This explains why the high-frequency improvements translate into a modest but consistent Top-1 gain.
>
> ---
> ## **3. Additional Experiments (weakness 3, Q3, Q5)**
> Thanks for the suggestions, we conducted additional expeiments to address the reviewer's concern.
> - **CUB-100.** We conducted the experiments on CUB-100 with GmNet-S1. The results is competitive which is shown below:
> | Variant | Params (M)  | Top-1 Acc (%) |
> |-|-|-|
> | ShuffleNet-V2  | 3.5  | 76.0 |
> | GmNet-S1| 3.1  | 81.5  |
> Compared to ShuffleNet-V2, GmNet-S1 achieves better performance with smller model size.
> - **Performance of changing the ReLU6 to ReLU.** Table 2 shows the performance difference when simply changing the ReLU6 to ReLU for GmNet-S3. Here we show the results of other variants in the following table:
> | Variant | ReLU6 → ReLU  |
> |-|-|
> | GmNet-S1| 75.5 → 74.7 |
> | GmNet-S2| 78.3 → 77.4 |
> | GmNet-S3| 79.3 → 78.3 |
> | GmNet-S4| 81.5 → 80.5 |
> -**Adapting GmNet in to RepVit/LSNet.** Adapting the GmNet to transformer based architectures for improving the performance is not straightforward since those model’s designs are more complex than GmNet. We mainly focus on efficient CNN-based architectures. The experiments of adapting GmNet to transformer-based architectures are meaningful but it’s out of the scope of our paper.
> ---
> ## **4. Unclear statements (Q1,2,4,6,7)**
> Thanks for pointing out the unclear statements in the paper. We will revise those in the following revision.
> - **Dataset of Fig3/4.** As mentioned in Sec.3, results in Fig3/4 are conducted on CIFAR-10.
> - **Confusion of convergence statements.** The goal of Fig. 3/4 is not to claim substantially faster overall convergence, but to illustrate **how different activations begin learning different frequency components at different stages of training**. Thus, these epoch-wise plots serve as a tool rather than a measure of final convergence speed.
> We agree that the wording “the convergence is faster” can be misleading, since the rightmost curves show similar final trends. We will revise the text to clarify that. This clarification does not affect the main results; it only provides intuition for how smooth and non-smooth activations emphasize different frequency ranges.
> - **Confusion on Fig.4.** Thank you for pointing this out. This is indeed a typo—Fig. 4 uses the threshold $r=10$, while results for other frequency thresholds are provided in Fig.13 in the supplementary. We will correct the text accordingly in the revised version.
> - **Justification of simplest GLU works best.** Since GmNet is a lightweight model, we mainly care about the accuracy, model size and latency, Tab 4/8 show that the simplest GLU achieves the best performance, smaller model size and the lowest latency.
> - **Typo.** Thank you for pointing this out. This is indeed a typo. We will correct the text accordingly in the revised version.

---

> > ### Comment · Reviewer_KXHh · 2025-11-25
> > **Thank you to the authors**
> >
> > Thank you for the very detailed rebuttal and for providing the additional experiment on CUB.
> >
> > However, I remain unconvinced about the connection between the shallow analysis in Section 3 and the observed performance gains. In fact, as reviewer i7B2 points out correctly, that the evidence in Table C suggests that the majority of the improvements stem from the 7×7 depthwise convolution. This is not necessarily a problem in itself, but it must be clearly acknowledged and discussed in the main paper. As it stands, the current analysis feels even less relevant—and potentially somewhat misleading.
> >
> > At this point, I am not inclined to increase my score. That said, the paper does have merit, and I am also not inclined to lower it below borderline accept.

---

> > > ### Author Response · Authors · 2025-11-25
> > >
> > > Thank you for the constructive feedback! We appreciate the reviewer’s concern regarding the connection between the analysis in Sec. 3 and the performance gains. In the final version, we will revise the main paper to more clearly acknowledge the dominant contribution of the 7×7 depthwise convolution. And we sill also emphasize that the gate is the key to break the bottleneck but it still needs a sufficient capability of capturing local information to make it work. We also thank the reviewer for recognizing the merits of the work, and we appreciate the thoughtful evaluation. If you have additional questions or concerns, please let us know.

---

### Official Review · Reviewer_HTpE · 2025-10-30

**Soundness:** 3
**Presentation:** 3
**Contribution:** 3
**Rating:** 8
**Confidence:** 2

**Summary:**

The paper introduces a theoretical study and experimental evaluation of Gated Linear Units (GLU) in CNNS from a Fourier perspective. In the theoretical part, the authors analyze the effects of non-linear activations of different variants of  ReLUs and GELUs and their properties in frequency space and their effect on the ability of networks to learn functions with high frequency components. Subsequently, the paper derives a theoretical analysis of GLUs in the same context by deriving the frequency mixing effect of element-wise multiplications in the frequency domain from the Convolution Theorem. This leads to the theoretical conclusion, that GLUs are suitable to allow networks to learn to preserve higher frequency information.

The paper then implements GMNet, a GLU enhanced version of MobileNetV2 and shows superior performance on ImageNet in comparison to SOTA efficient image classification networks. The authors conclude, that the effect of GLUs is most beneficial efficient network architectures (for mobile devices), which are known to have a stronger in low-frequency bias.

**Strengths:**

The paper tackles an important theoretical question: the properties of simple network components liker activation functions and element-wise multiplications (and a such GLUs) have not been studied even though these properties of other components (like convolution filters, padding and down-sampling units) have shown significant impact on the performance and robustness/generalization abilities of CNNs.

On the practical side, the introduced GMNet confirms the theoretical analysis and shows significant performance increases over efficient SOTA architectures.

**Weaknesses:**

N1: The introductory part is hard to read for someone who is not working on related problems. Especially the difference between the frequency representation of inputs (read data and feature maps in later layers) and the the frequency representation of the learned decision function is not well explained and leaves unclarities to which the the low-frequency bias is actually relating.

N2: Also the introductory and related work section somewhat neglect the bigger picture of similar studies conducted on other components (especially convolutions and down-sampling) - for example [1] and [2]. While some works are cited, the relation of these works are not put into perspective.

N3: as the authors mentioned, basing decisions increasing on high frequencies, can result in aliasing [3] and significant loss of robustness [2]. Here an ablation study would be very beneficial.

N4:  the experimental evaluation is limited to low resolution ImageNet data. This also directly limits the frequency representation. It would be good to see how the proposed approach behaves on high resolution data.

**Minor:**
* Figure 1 is not matching it's caption: there are only two different frequency inputs (not three) and where is that frog? Also the schematic illustration of fig 1 does not actually show the effect of GLUs (on real data)-> it actually would be nice to have a plot like this: real data -> ReLU vs GELU vs GLU output in spatial and frequency domain.


[1] Durall et. al. "Watch your up-convolution: Cnn based generative deep neural networks are failing to reproduce spectral distributions." Proceedings of the IEEE/CVF conference on computer vision and pattern recognition. 2020.

[2] Grabinski, et al. "Frequencylowcut pooling-plug and play against catastrophic overfitting." European Conference on Computer Vision. Cham: Springer Nature Switzerland, 2022.

[3] Grabinski, Julia, Janis Keuper, and Margret Keuper. "Aliasing and adversarial robust generalization of cnns." Machine Learning 111.11 (2022): 3925-3951.

**Questions:**

Q1:  I'm curious what the GLUs are actually learning. Vitalization of the GLU weight by layer and the resulting changes in the frequency spectrum would be very informative. It would be very interesting to know in which parts of the network higher frequency information is relevant

Q2: the evaluation focuses on efficient nets. it would be very interesting to see GLUs in large modern CNNs like ConvNext (v2). I would speculate, that the larger kernels in large CNNs (which are less band limited) mitigate the effects of GLUs. This would not harm the results of this paper, but would further increase the general insights. Have the authors tried this?

---

> ### Author Response · Authors · 2025-11-22
>
> We sincerely appreciate the reviewer's recognition of our work. Your thoughtful feedback has helped us improve the clarity and rigor of the work. Here is our response:
>
> ---
> ## **1. Frequency representation in introduction**
> We thank the reviewer for pointing out this clarity issue. In the introduction, our use of “frequency representation” refers specifically to the **frequency components of the input signal.** That is, the notion of low-frequency bias we discuss relates to **how the network responds to low- vs. high-frequency components in the input，**and how different activation functions modulate these components as they propagate through the convolutional layers. We are *not* referring to the frequency characteristics of the **learned decision function** or classifier boundary.
>
> We appreciate the reviewer’s feedback and will revise the introduction to explicitly distinguish:
>  (1) input feature frequencies,
>  (2) internal feature spectra, and
>  (3) the frequency properties of the classifier function,
>  to avoid ambiguity and improve readability.
>
> ---
> ## **2. Neglecting the bigger picture of similar studies**
> We thank the reviewer for pointing out this broader context. We will revise the introduction and related work to better position our contribution within this broader landscape of frequency-domain analyses and explicitly discuss how our findings relate to frequency behaviors caused by convolutions and sampling operators.
>
> Several studies have demonstrated that architectural components such as  up-convolutions, strided convolutions, and pooling introduce characteristic frequency distortions in CNNs [1, 2].
> These works highlight the importance of understanding frequency behavior at the  level of spatial resampling operations. In contrast, our study focuses on the spectral effects induced by the gating mechanisms and activation functions inside lightweight CNN blocks. We will add those discussions in the related work section of the following revision.
>
> ---
> ## **3. Studies on aliasing and robustness**
> Following the reviewer’s suggestion, we train Res18-Gate-ReLU and Res18-Gate-GELU on CIFAR-10 which are defined in Fig.2 following the setting of \[2\] to show how different activation functions affect the robust.
> We conducted an adversarial robustness ablation under PGD attacks. As shown below, the ReLU-based GLU—which emphasizes higher-frequency components—indeed exhibits slightly lower PGD accuracy:
> * **Res18-Gate-ReLU:** 45.7% PGD accuracy
> * **Res18-Gate-GELU:** 46.7% PGD accuracy
>
> Clean accuracy remains similar for both variants (82.2% vs. 82.8%), but the \~1% drop in PGD robustness for the ReLU gate is consistent with prior findings that stronger high-frequency reliance increases the influence of aliasing and adversarial perturbations. We will include this ablation and brief discussion in the revised paper.
>
> ---
> ## **4. Expeiments on higher resolution data**
> Thanks for the suggestions, we conduct experiments on training GmNet-S4 with a higher resolution input. Increasing the image size from 224 to 384. It further increases the network's capacity and accuracy with more richer frequency singals.
> ### Table A. Results of increasing the input resolution.
> | Model | Top-1 Acc (%) under $224^2$ | Top-1 Acc (%) under $384^2$ |
> |-|-|-|
> | MobileNetv4-Conv-M | 79.9 | 80.5 (+0.6)|
> | GmNet-S4  | **81.5**|**82.6 (+1.1)**|
> Comapred to MobileNetv4-M, our model has larger improvements which indicates that our glu-based architicture has better potential learning with higher resolution datasets.
>
> ---
> ## **5. Spectral evidence**
> Good point! To address this, we performed an additional layer-wise spectral study—pre-/post-gate FFT comparisons, and high/low-frequency energy ratios—please refer to our response to **the first question of Reviewer i7B2**. These analyses reveal how the GLU modulates different frequency components at different depths. We will incorporate a concise summary of these findings and the visualizations into the revised manuscript to improve clarity.
>
> ---
> ## **6. Experiments on ConvNext**
> To address the reviewer’s concerns, we adapted our GLU on ConvNeXt-Tiny and trained the model with the same setting as GmNet on ImageNet-1k. As shown in Table B, GLU could improve the performance without any additional costs.
>
> The improvement is not as significant as that in GmNet which might be caused by the large CNNs are less affected by activation-induced frequency bias. The block should be designed specifically based on the different architecture. The reviewer proposed a good point. We will add this observation to the revised manuscript.
> ## Table B. ConvNext vs. ConvNext+GLU
> |Model|Top-1 Acc (%)|Params (M)| FLOPs (G) |
> |-|-|-|-|
> |ConvNext| 82.5| 28.6 | 4.46 |
> |ConvNext + GLU| **83.0**| 28.6 | 4.46|
>
> [1] Durall et. al. "Watch your up-convolution: Cnn based generative deep..." CVPR. 2020.
>
> [2] Grabinski, et al. "Frequencylowcut pooling-plug and ...."ECCV, 2022.

---

> > ### Comment · Reviewer_HTpE · 2025-11-24
> >
> > Thank you for your extensive response which answers my questions. Looking at the other reviews, I think the authors did a good job to resolve the points raised there. Hence, I'll argue in favor of acceptance of the paper and would ask my may fellow reviewers to consider raising their scores.

---

> > > ### Author Response · Authors · 2025-11-24
> > >
> > > Thank you very much for your follow-up and for the encouraging remarks! We truly appreciate the time you put into reviewing our response, as well as your willingness to advocate for its acceptance. Let us know if you have any additional questions.

---

### Official Review · Reviewer_i7B2 · 2025-10-30

**Soundness:** 2
**Presentation:** 3
**Contribution:** 3
**Rating:** 6
**Confidence:** 4

**Summary:**

This paper reinterprets GLU gating from a frequency-domain standpoint and derives design rules that are instantiated in a compact backbone called GmNet. Concretely, the model uses a shared-representation gate, 7×7 depthwise convolutions to mix frequency bands, and favors ReLU6 for a more stable high/low-frequency trade-off. The authors also propose a DFT-based frequency-split evaluation protocol and run ablations on activation choice, gating variants, and latency–accuracy trade-offs. Experiments show GmNet achieves good performance on ImageNet-1K, with runtime comparisons against efficient baselines. Evidence is provided mainly on ImageNet classification. Overall, the primary contribution is an explanatory frequency view that motivates a modest architectural tweak for efficient vision models, supported by single-task experiments and ablations.

**Strengths:**

1. Recasts GLU with a frequency-domain hypothesis and condenses it into a single, drop-in GmNet block (shared-representation gate + flanking 7×7 depthwise convs, ReLU6), keeping parameters/latency overhead small and integration into lightweight CNNs straightforward.
2. On ImageNet-1K, reports SOTA/near-SOTA accuracy–latency trade-offs against strong efficient baselines under matched settings, with consistent gains across model scales.

**Weaknesses:**

1.	The frequency claim remains qualitative and no per-layer power spectra, gain/attenuation curves, or controlled sinusoid/wavelet probes; “high-frequency amplification” is hard to verify quantitatively.
2.	Architectural novelty is modest (GLU + 7×7 DWConv + ReLU6); gains may conflate with kernel size/width/depth/recipe. Lacks a contribution breakdown under strictly matched FLOPs/params and training budgets.
3.	Evidence is centered on ImageNet in the main text; downstream/cross-dataset/robustness results are relegated to the supplement and explored shallowly, limiting generality.
4.	Measurement/reporting gaps: mostly single-run top-1 without confidence intervals; latency lacks a standardized, reproducible protocol (hardware, batch size, warm-up, operator fusion).
5.	Positioning vs contemporaneous gating/frequency methods is under-specified.

**Questions:**

1.	Would you be able to provide quantitative spectral evidence—e.g., per-layer power spectra before/after gating, high/low-frequency energy ratios, and responses to controlled sinusoid/wavelet probes—and also formulate a clear, testable proposition about the gate’s frequency response that could, in principle, be falsified by these experiments?
2.	Under strictly matched FLOPs, parameter counts, and training recipes, could you report a contribution breakdown for (a) baseline, (b) +7×7 depthwise conv only, (c) +gate only, (d) +ReLU6 only, and (e) full GmNet, including mean ± std over at least three random seeds?

---

> ### Author Response · Authors · 2025-11-22
>
> We sincerely appreciate the reviewer's recognition of our work and your thoughtful feedback, which has helped us improve the clarity and rigor of the work. Here is our response:
>
> ---
> ## ** 1. Quantitative spectral evidence **
> Good Point! To address the reviewer’s request, we computed *high/low-frequency energy ratios* across multiple layers and model variants. Table A shows the spectral changes from the layer *before* to the layer *after* the gate (f1 → g), demonstrating that the gate indeed reshapes the frequency distribution. To illustrate how different activation functions affect the model’s frequency response, we compute the high/low-frequency energy ratios of the first 7×7 dwconv layers in each stage (Table B), defining low frequencies as the central 1/4 region of the 2D spectrum.
>
> ## Table A. High/Low Frequency Ratio Comparison of f1→g
> | Stage | Layer Pair | ReLU6 (H/L) | GELU (H/L) | ReLU (H/L) |
> |-|-|-|-|-|
> |0.1| f1 → g| 0.11954 → 0.11997 | 0.0575 → 0.00398 | 0.51720 → 0.56740 |
> |1.1| f1 → g| 0.09888 → 0.14288 | 0.04231 → 0.00224 | 0.14221 → 0.17506 |
> |2.1| f1 → g| 0.03857 → 0.07056 | 0.02515 → 0.00130 | 0.00182 → 0.00301 |
> |3.1| f1 → g| 0.00193 → 0.02809 | 0.01063 → 0.00030 | 0.00315 → 0.03407 |
> **Observation:**
> - **GELU**: f1 → g often *increases* low-frequency response
> - **ReLU6 / ReLU**: f1 → g *increases* the high-frequency ratio.
>
> ## Table B. High/Low Frequency Ratio Comparison at DWConv
> | Stage | GELU (H/L) | ReLU (H/L) | ReLU6 (H/L) |
> |-|-|-|-|
> | 0.1   | 0.1203 | **1.1553** | 0.7057 |
> | 1.1   | 0.0695 | **0.3040** | 0.2452 |
> | 2.1   | 0.0381 | 0.0024 | **0.0697** |
> | 3.1   | 0.0057  | 0.0088 | **0.0142** |
> **Observation:**
>
> **Stage 0 & 1:** ReLU exhibits the strongest high-frequency response in the early stages, consistent with its non-smooth activation behavior and its tendency to preserve or amplify sharp signal transitions in early layers.
> **Stage 2 & 3:** ReLU6 shows the highest high/low ratio in the later stages, suggesting that its clipped nonlinearity becomes more influential in deeper layers which may indicate why the model using ReLU6 delivers the best overall performances.
> **Across all stages:** GELU consistently produces the lowest high/low ratios, confirming its smoothness-induced preference for low-frequency features.
>
> These results are stable across blocks and models and fully consistent with our hypothesis that smooth activations (e.g., GELU) bias the network toward low-frequency representations, while non-smooth activations (ReLU/ReLU6) amplify high-frequency components. This constitutes a *falsifiable* prediction: if a smooth activation were to produce systematically *higher* high/low ratios than ReLU/ReLU6 across comparable layers, our hypothesis would be invalidated. The spectral evidence presented here therefore directly supports the intuitive explanation in Sec 3.2.
>
> ---
> ## **2. Contribution breakdown**
> Following the reviewer’s request, we show the ablation study under **strictly matched FLOPs, parameter count, and training settings**, and report the mean ± std over three random seeds. We replace the *7×7 dwconv* with *a linear layer* and replace *the GLU* with *a ReLU* as the baseline block.  The results are shown in Table C.
> ## Table C. Contribution breakdown.
> | Variant | Params (M) | FLOPs (G) | Top-1 Acc (%) |
> |-|-|-|-|
> |Baseline| 7.82 | 1.24 | 71.5 ± 0.2 |
> |+7×7 DWConv only| 7.82 | 1.28 | 77.8 ± 0.1 |
> |+Gate (Identity)| 7.82 | 1.24 | 69.2 ± 0.3 |
> |+ReLU6 only| 7.82 | 1.24 | 77.5 ± 0.1 |
> |Full GmNet| 7.82 | 1.24 | **79.2 ± 0.1** |
> Overall, Table C shows the contribution of each design where 7×7 DWConv contributes most sololy. It also indicates that a larger kernal is need to capture the local relationship to make the GLU effective. Otherwise, amplifying certain frequency signals in a blindly way is even harmful due to the limited model size.
>
> ---
> ## **3. Downstream tasks**
> Thanks for the suggestion, we have done experiments on object detection, instance/semantic segmentation in Table 9. We will move the downstream results to the main paper in the following revisions.
>
> ---
> ## **4. Measurement protocol**
> We evaluate the latency on A100-40G GPU and iPhone14 mobile through all models. We set the batch size to $1$. We apply a 50 iters warm-up and report the average over 400 iters. All results are measured in eager mode with ONNX benchmark without operator fusion.
>
> ---
> ## **5. Positioning**
> Our method is different from prior works which focus on channel recalibration but do not analyze their spectral effects. Our contribution lies in showing that the glu-based gating inside lightweight CNNs itself introduces a systematic frequency influence, which has not been characterized in existing works. We will update the introduction and related work to position GmNet more clearly relative to these concurrent lines of work.

---

> > ### Comment · Reviewer_i7B2 · 2025-11-24
> > **I will maintain my initial rating of 6.**
> >
> > I sincerely thank the authors for the comprehensive and rigorous rebuttal, particularly for providing the requested quantitative evidence in Tables A, B, and C. The spectral analysis Tables A & B convincingly supports the hypothesis that non-smooth activations bias the network toward higher frequencies, consistent with signal processing theory.
> >
> > However, regarding Table C, this evidence unequivocally demonstrates that the adoption of the large 7×7 kernel is the primary driver of performance, contributing approximately 82% of the total gain 6.3% out of a total 7.7% improvement. The Gating Mechanism and the Frequency View—the elements emphasized in the paper's title and abstract—only provide a marginal refinement.
> >
> > While the synergy between the large kernel to build the base features and the frequency-aware gate to refine them is valuable, the paper's current framing is somewhat misleading. Positioning the paper as Revisiting Gating Mechanisms From A Frequency View when the 7×7 kernel is the main workhorse raises a concern about whether the narrative accurately reflects the core contribution.
> >
> > Given the strong final empirical results and the detailed analysis provided, I will maintain my initial rating of 6.

---

> > > ### Author Response · Authors · 2025-11-24
> > >
> > > Thanks very much for your follow-up and the encouraging remarks! We truly appreciate the time you put into reviewing our response and the thoughtful assessment.
> > >
> > > Thanks for highlighting the relative contribution of the 7×7 depthwise kernel in Table C. We agree that sufficiently large local receptive fields are necessary for enabling GLU to operate effectively. In this sense, the 7×7 kernel provides the essential base capacity for capturing local frequency patterns.
> > >
> > > However, our core claim is that **the gate is what allows the network to break the bottleneck**. While the 7×7 kernel contributes the largest portion of the raw improvement over the baseline, the GLU mechanism consistently delivers the additional margin that **lifts the model to state-of-the-art performance among lightweight architectures**—without introducing significantly computational cost. This is precisely why we emphasize the value of the gating mechanism within the lightweight-model regime.
> > >
> > > Again, we sincerely appreciate the reviewer’s time and constructive suggestions. If you have additional questions or suggestions. Please let us know.

---

### Official Review · Reviewer_krko · 2025-10-31

**Soundness:** 3
**Presentation:** 3
**Contribution:** 2
**Rating:** 4
**Confidence:** 5

**Summary:**

This paper introduces GmNet, a new architecture for lightweight Convolutional Neural Networks (CNNs) that achieves state-of-the-art performance. The core idea is the adaptation of the Gated Linear Unit (GLU) mechanism, which has been highly successful in language models, to the domain of efficient computer vision. The authors present a systematic analysis of GLU from a frequency perspective, demonstrating that the element-wise multiplication within GLU acts as a convolution in the frequency domain, which can selectively amplify high-frequency signals. Based on these insights, they introduce the Gating Mechanism Network (GmNet), a simple and efficient architecture that incorporates these frequency-aware gating principles. Without complex training strategies, GmNet sets a new state-of-the-art for efficient models, with the GmNet-S4 variant achieving 81.5% top-1 accuracy on ImageNet-1K, validating the effectiveness of designing lightweight models that can learn from a full spectrum of frequencies.

**Strengths:**

- This paper shows not only the number of FLOPs and parameters, but also the model latency on both GPU and a mobile device, which demonstrates the effectiveness of the proposed lightweight model.
- The proposed GmNet block is simple and clean, integrating the gating unit effectively without excessive architectural complexity. But the state-of-the-art (SOTA) empirical results on the ImageNet benchmark show its effectiveness.
- The authors perform thorough ablation studies that validate their design choices. They compare different activation functions and different GLU designs.

**Weaknesses:**

- The novelty of the proposed method is limited. In the current LLM design, GLU variants are widely used in the gated MLP modules. Those LLM models already demonstrated the effectiveness of those designs.
- In CNNs, many "quasi-attention" methods, e.g., SENet[1], GCNet[2], have already explored the element-wise multiplication of the activation tensors, and their results show that the gated design in CNNs can help achieve better model accuracy, similar to the findings of this GmNet. It would be better to compare GmNet with those methods to demonstrate its advantages.
- The results in Table 6 cannot support the hypothesis in Section 3.2 (or Figure 4) that ReLU6 can be better for high-frequency data and not good for low-frequency data. The accuracy gain should mainly come from the activation function design, which leads to better model performance in general. This improvement doesn't seem to be related to how the activation function processes low or high frequencies.

[1] Jie Hu, et al. Gather-Excite: Exploiting Feature Context in Convolutional Neural Networks, NeurIPS, 2018

[2]Yue Cao, et al. GCNet: Non-local Networks Meet Squeeze-Excitation Networks and Beyond, ICCV workshop, 2019

**Questions:**

- The swiGLU and SiLU are commonly used in the gated MLP of current LLMs design, and many of those LLMs have demonstrated the effectiveness of these activation functions. It would be better to evaluate these two activations in CNN models to see whether they can also improve model accuracy.
- Typos:
	- L45: showed -> shown
	- L318: traini ng -> training
	- L427: Moverover -> Moreover
	- L86: duplicated sentences:
		- Consequently, our approach is inherently more effective in preserving and enhancing high-frequency information.
		- As a result, the former structure is inherently more effective in preserving and enhancing high-frequency information.

---

> ### Author Response · Authors · 2025-11-22
>
> We sincerely appreciate the reviewers’ thoughtful feedback, which has helped us improve the clarity, rigor, and presentation of the work. Here is our response:
>
> ---
> ## **1. The novelty of the proposed method.**
> We appreciate the reviewer’s comment regarding the widespread use of GLU variants in modern LLM architectures. However, our work targets a fundamentally different problem setting and contributes novelty along dimensions that have not been explored in prior literature.
>
> Unlike LLMs, which employ GLUs primarily as *effective black-box gating units* within *large-scale transformer-based MLPs*, our study focuses on **visual tasks with efficient and lightweight convolutional models**. The computational budgets, spatial priors, and feature distribuition differ substantially from those in language models.
>
> Our contribution is not only showing that “GLU works.” Instead, we provide
> - (1) **A systematic investigation** of why gating can benefit efficient CNNs—revealing the interaction between the element-wise multiplication and different activation functions;
> - (2) a principled, lightweight GLU-based module specifically designed for **lightweight vision backbones**, which is different in structure, motivation, and deployment advantages compared to the GLUs used in LLMs.
>
> To our knowledge, no prior work has established such an analysis-driven design pipeline or demonstrated that GLU-based gating can be optimized for lightweight CNNs to achieve consistent performance gains.
> Therefore, as we addressed in the introduction and related work, while GLUs are indeed present in LLM literature, **our novelty lies in interpreting and tailoring these ideas to the lightweight vision network domain.**
>
> ---
> ## **2. Comparison with quasi-attention methods**
> We thank the reviewer for the suggestion. To address this comment, we implemented a GE-based channel-recalibration block based on \[1\] on GmNet-S3 and ran experiments under identical settings. The results show that the GE-based variant improves slightly, but **our GmNet block still achieves higher accuracy with fewer FLOPs and number of parameters**, demonstrating its advantage over existing quasi-attention designs.
>
> ### Table A. Comparison between GE-based GmNet and GmNet-S3.
> | Model  | Top-1 Acc (%) | FLOPs (G) | Params (M) |
> |-|-|-|-|
> | GE-based GmNet    | 74.4  | 1.5  | 8.4  |
> | GmNet-S3  | **79.3**  | 1.2  | 7.8    |
>
> Moreover, methods such as GENet/GCNet rely on **global pooling**, which removes local high-frequency information and produces coarse channel weights. In contrast, GmNet performs **spatially local and content-dependent gating** through depthwise convolutions and gating interactions. This design preserves high-frequency details and aligns better with the needs of lightweight CNNs.  Prior works also do not analyze this mechanism, whereas our study provides both **experimental insight** and **a tailored block design** for efficient vision models.
>
> ---
> ## **3. Relationship between the frequency response and the overall performance**
> We assume the reviewer is referring to Table 2 instead of Table 6 because the Table 6 is about the training setting. This table indeed supports our hypothesis: **the ReLU6-based GLU shows consistently higher accuracy on high-frequency components** across all threshold settings, whereas its performance on low-frequency components **can not surpass other models consistently**. For example, even though the ReLU6 model achieves the best overall accuracy, its low-frequency accuracies at $r=36$ and $r=48$ are still **lower than that of GELU**, confirming that ReLU6 is less effective in capturing low-frequency signals.
>
> Thus, the observed improvement is **not solely due to a generally “better activation function”**, but is closely aligned with the frequency-selective behavior analyzed in Section 3.2 and visualized in Figure 4\. The strong gains on high-frequency features naturally contribute more to the overall accuracy in our efficient CNN setting, and this observation is fully consistent with our hypothesis.
>
> ---
> ## **3. Performance with swiGLU and SiLU**
> Thanks for pointing that out. We have conducted experiments with the SiLU variant on CIFAR-10 to show the learning dynamic in Fig. 14. Following the reviewer’s advice, we adapt both SwiGLU and SiLU in GmNet-S3 and train them under the same training settings. The results in Table B show that both activations provide improvements compared to the baseline which removes the activations in the GLU. Their gains are consistently smaller than those achieved by our proposed design, confirming that GmNet is better aligned with the frequency characteristics and architectural constraints of lightweight models.
> ## Table B. Reults of using SiLU/SwiGLU.
> | Model / Activation  | Top-1 Acc (%) |
> |-|-|
> | Baseline (Identity)| 70.5 |
> | + SiLU |77.9  |
> | + SwiGLU | 77.2  |
> | + Proposed (ReLU6-GLU)  | **79.3** |
>
> ---
> ## **4. Typos**
> Thanks for your careful reading. We will fix typos in the revision.

---

> > ### Comment · Reviewer_krko · 2025-11-26
> >
> > Thank you for the detailed response and the additional experiments. This addressed most of my concerns. But after reviewing 'Table C. Contribution breakdown' and the other reviews, I agree with Reviewer i7B2's point that the performance improvement is mainly driven by the 7x7 DW convolution, not the frequency-domain methods. Since this benefit is unrelated to the frequency-domain motivation emphasized in the title, I will maintain my current rating.

---

> > > ### Author Response · Authors · 2025-11-27
> > >
> > > Thanks very much for your follow-up and the encouraging remarks! We truly appreciate the time you put into reviewing our response and the thoughtful assessment.
> > >
> > > We have updated addtional experiments on the top for addressing your concerns.
> > >
> > > Again, we sincerely appreciate the reviewer’s time and constructive suggestions. If you have additional questions or suggestions. Please let us know.

---

### Author Response · Authors · 2025-11-27
**Additional experiments on the contribution breakdown**

Dear Reviewer krko, Reviewer i7B2 and Reviewer KXHh,

Thank you for your insightful reviews and feedback. We have noticed that reviewers are bringing attention to Table C in our response to Reviewer i7B2. Inspired by your comments, we performed additional experiments to further clarify the contribution of the gating mechanism relative to the 7×7 depthwise convolution.

We revisited the *gate-only* variant in Table C and realized that it does not represent a functional gating mechanism. The use of an identity gate is incomplete and can be misleading, as the gate without any activation function significantly affects performance (consistent with observations in Table 2 of the main paper). To properly reflect the behavior of a gating module, we evaluated variants that incorporate different activation functions while keeping parameters and FLOPs matched. The results are shown below.

## Table A. Contribution breakdown under matched FLOPs/Params (mean ± std).
| Variant | Params (M) | FLOPs (G) | Top-1 Acc (%) |
|---------|------------|------------|----------------|
| (a) Baseline             | 7.82 | 1.24 | 71.5 ± 0.2 |
| (b) +7×7 DWConv only     | 7.82 | 1.28 | 78.1 ± 0.1 |
| (c) +Gate (Identity)  only   | 7.82 | 1.24 | 69.2 ± 0.3 |
| (d) +Gate (ReLU) only    | 7.82 | 1.24 | 78.0 ± 0.2 |
| (e) +Gate (GELU)  only   | 7.82 | 1.24 | 77.9 ± 0.1 |
| (f) +Gate (ReLU6)  only   | 7.82 | 1.24 | 78.5 ± 0.1 |
| (g) +ReLU6 only          | 7.82 | 1.24 | 77.9 ± 0.1 |
| (h) Full GmNet           | 7.82 | 1.24 | **79.2 ± 0.1** |

As shown, once functional activations are used, **the gating variants no longer underperform and are largely comparable to the 7×7 DWConv-only model, with the ReLU6 gate performing better.** This demonstrates that the earlier weak result arose from the identity formulation rather than from an inherent weakness of gating. **These results also indicate that the performance gains introduced by gating are not marginal; instead, the gating mechanism provides a distinct improvement beyond what the large kernel alone can offer.**

We will revise the paper to clarify this point and avoid any ambiguity in the framing. We sincerely appreciate the reviewers’ comments, which helped us substantially improve this part of the analysis.

---

### Author Response · Authors · 2025-12-03
**Summary of the discussion period**

Dear ACs,

We thank the Area Chairs for their time and effort in overseeing the discussion process. Based on the exchanges with the reviewers, we believe that our responses have sufficiently addressed the main concerns. In particular, Reviewer i7B2 raised a question regarding the interpretation of Table C in our response to the Reviewer i7B2, which is related to the relative contributions of the 7×7 depthwise convolution and the gating mechanism. To clarify this point, we conducted additional experiments with fully functional gating variants. These results show that gating provides a distinct and meaningful improvement beyond what the large kernel alone can achieve.

Following the reviewers’ feedback, we have revised the paper and uploaded an updated version with all modifications highlighted. The revision corrects typos and ambiguous phrasing, and includes additional experiments and analysis that directly address the reviewers’ comments.

We sincerely appreciate the constructive feedback from all reviewers, which has significantly strengthened the paper.

Best regards,

Authors

---

### Meta-Review · Area_Chair_c9V5 · 2026-01-03

**Summary:**

This paper reinterprets GLU gating from a frequency-domain standpoint and derives design rules that are instantiated in a compact backbone called GmNet. It gets 4, 6, 8, 6 in the first round. The main concerns are limited novelty, claims, theoretical depth, writting problems. In rebuttal, most of these concerns are addressed except 7×7 depthwise convolution and motivation emphasized in the title. I am leaning to accept this paper. Author should revise the paper according to discussion.

**Reviewer Concerns:**

Most of reviewers concerns are addressed except 7×7 depthwise convolution and motivation emphasized in the title.

**Reviewer Scores:**

Reviewer krko would not change their score.
Reviewer i7B2 would not change their score.
Reviewer HTpE would not change their score.
Reviewer KXHh would not change their score.

---

### Decision · Program_Chairs · 2026-01-26

Accept (Poster)